# Distinct mechanisms mediate speed-accuracy adjustments in cortico-subthalamic networks

**Damian M Herz[1,2], Huiling Tan[1,2], John-Stuart Brittain[1,2], Petra Fischer[1,2], Binith Cheeran[2], Alexander L Green[2], James FitzGerald[2], Tipu Z Aziz[2], Keyoumars Ashkan[3], Simon Little[4], Thomas Foltynie[4], Patricia Limousin[4], Ludvic Zrinzo[4], Rafal Bogacz[1,2], Peter Brown[1,2]***

[1]Medical Research Council Brain Network Dynamics Unit at the University of Oxford, Oxford, United Kingdom; [2]Nuffield Department of Clinical Neurosciences, John Radcliffe Hospital, University of Oxford, Oxford, United Kingdom; [3]Department of Neurosurgery, King's College Hospital, London, United Kingdom; [4]Unit of Functional Neurosurgery, Sobell Department of Motor Neuroscience and Movement Disorders, University College London Institute of Neurology, London, United Kingdom

**Abstract** Optimal decision-making requires balancing fast but error-prone and more accurate but slower decisions through adjustments of decision thresholds. Here, we demonstrate two distinct correlates of such speed-accuracy adjustments by recording subthalamic nucleus (STN) activity and electroencephalography in 11 Parkinson's disease patients during a perceptual decision-making task; STN low-frequency oscillatory (LFO) activity (2–8 Hz), coupled to activity at prefrontal electrode Fz, and STN beta activity (13–30 Hz) coupled to electrodes C3/C4 close to motor cortex. These two correlates differed not only in their cortical topography and spectral characteristics but also in the relative timing of recruitment and in their precise relationship with decision thresholds. Increases of STN LFO power preceding the response predicted increased thresholds only after accuracy instructions, while cue-induced reductions of STN beta power decreased thresholds irrespective of instructions. These findings indicate that distinct neural mechanisms determine whether a decision will be made in haste or with caution.

**\*For correspondence:** peter.brown@ndcn.ox.ac.uk

**Competing interests:** The authors declare that no competing interests exist.

## Introduction

Fast decisions come at the cost of reduced accuracy. This elementary aspect of decision-making, often referred to as speed-accuracy trade-off, has been studied for over a century (for a review see *Heitz, 2014*) and can be observed in a multitude of tasks and across various species including rats, non-human primates and humans (*Bogacz et al., 2010*; *Forstmann et al., 2010*, *2008*; *Hanks et al., 2014*; *Heitz and Schall, 2012*; *Ivanoff et al., 2008*; *Ratcliff and McKoon, 2008*, *1998*; *Reinagel, 2013*; *Schouten and Bekker, 1967*; *Thura and Cisek, 2016*; *van Veen et al., 2008*; *Wickelgren, 1977*). For example, when approaching prey, the hunter must plan the advance in order to avoid mistakes. However, if deliberation takes too long, the prey might escape and the attempt would be to no avail. Therefore, it is crucial for intelligent agents to balance opposing demands of speed and accuracy.

Mathematically, speed-accuracy adjustments can be implemented by elevating or lowering the 'decision threshold', that is, a criterion which defines when the continuous accumulation of evidence

**eLife digest** In everyday decisions, we have to balance how quickly we need to make a decision with how accurate we want our decision to be. For example, if you plan your next holiday you might want to make sure that you pick the best destination without caring too much about the time it takes to arrive at that decision. On the other hand, in your lunch break you might want to quickly choose between the different meals on the menu to make sure you are back at work on time, even though you might overlook a dish that you would have preferred. This effect – that decisions we make in haste are more likely to be suboptimal than slower, more deliberate decisions – is known as the speed-accuracy trade-off.

One theory suggests that the activity of a brain area termed the subthalamic nucleus reflects whether people will prioritize speed or accuracy during decision-making. This area is seated deep inside the brain, meaning that it is normally difficult to record its activity.

Herz et al. have now recorded the activity of the subthalamic nucleus in individuals with Parkinson's disease who underwent brain surgery as part of their treatment. When these individuals switched between fast and cautious decision-making, the activity in the subthalamic nucleus changed, as did its relationship with the activity seen in other brain areas. Furthermore, these activity changes predicted how much information participants acquired before committing to a choice.

Deep brain stimulation of the subthalamic nucleus is now a standard treatment for Parkinson's disease. It will be important to assess whether this treatment affects the changes in subthalamic activity that are related to decision-making, and whether this affects whether an individual is more likely to make fast or accurate decisions.

should be terminated and the option with the strongest evidence chosen (*Bogacz et al., 2006*; *Ratcliff and McKoon, 2008*). In neurobiological models of decision-making, such modulations of decision thresholds subserving speed-accuracy trade-offs have been assigned to the basal ganglia (BG) (*Bogacz et al., 2010*; *Frank, 2006*; *Lo and Wang, 2006*), which are connected to a wide range of decision- and movement-related cortical areas in a closed-loop fashion (*Alexander et al., 1986*; *Lambert et al., 2012*). Within these loops, the BG exert tonic inhibition over cortical areas, which can be decreased through activation of a facilitatory, direct pathway connecting the striatum with BG output areas or increased by activation of two net inhibitory circuits passing through the subthalamic nucleus (STN), which incorporate the indirect and hyperdirect pathways (*Alexander et al., 1986*; *Bogacz et al., 2010*; *Kravitz et al., 2010*). Increased activity of cortical neurons computing decision-related signals during speed emphasis, as predicted by this model, has been observed as firing rate changes in non-human primates (*Hanks et al., 2014*; *Heitz and Schall, 2012*; *Thura and Cisek, 2016*) and indicated by functional magnetic resonance imaging (fMRI) studies in humans (*Forstmann et al., 2008*; *Ivanoff et al., 2008*; *van Veen et al., 2008*). At the BG level, fMRI studies have inferred increased activity in striatum during speed emphasis (*Forstmann et al., 2008*; *van Veen et al., 2008*). An involvement of the STN has been suggested by studies showing that treatment with deep brain stimulation (DBS) of the STN affects patients' ability to switch between fast and accurate decision-making (*Green et al., 2013*; *Pote et al., 2016*). However, these behavioral studies are non-informative regarding the neurophysiological correlates of speed-accuracy adjustments.

To address this, we exploited the very high spatial and temporal resolution of local field potential (LFP) recordings directly from the STN while Parkinson's disease patients who had undergone STN DBS surgery performed a perceptual decision-making task. Simultaneous recordings of electroencephalography (EEG) at electrodes C3 and C4 over or close to the motor cortex, and electrode Fz over the prefrontal cortex as well as computational modeling of participants' latent decision-making parameters allowed us to relate modulations of STN activity and cortico-STN connectivity to adaptations of decision threshold during speed-accuracy adjustments.

# Results

## Behavior

Eleven patients with Parkinson's disease performed a moving dots task, in which they had to decide whether a cloud of moving dots appeared to move to the left or to the right. Task difficulty was manipulated by changing the percentage of dots moving coherently in one direction (8% or 50%). Before the onset of the moving dots, subjects were either instructed to respond as quickly or as accurately as possible (*Figure 1A*). While trials with low and high coherence were pseudorandomly interspersed, speed vs. accuracy instructions alternated in blocks of 20 trials (*Figure 1B*) resulting in a 2 (low vs. high coherence) * 2 (speed vs. accuracy instructions) design (*Figure 1C*). An overview of the behavioral data is given in *Figure 1D–E*. Analysis of reaction times (RT) showed that participants were significantly faster during high compared to low coherence trials (974 ± 302 ms vs. 1478 ± 298 ms, main effect of coherence: $F_{(1,10)}$ = 83.284, p<0.001) and when they were instructed to emphasize speed over accuracy (1146 ± 322 ms vs. 1306 ms ±259 ms, main effect of instruction: $F_{(1,10)}$ = 18.172, p=0.002). In addition, there was a significant interaction of instruction*coherence ($F_{(1,10)}$ = 9.744, p=0.011), since speed instructions led to a stronger decrease in RT during low coherence compared to high coherence (180 ± 137 ms vs. 77 ± 113 ms decrease in RT, $t_{(10)}$ = 2.662, p=0.024). This difference was, however, not significant when considering % change in RT rather than the absolute difference ($t_{(10)}$ = 1.414, p=0.188). Accuracy rates were significantly lower during low vs. high coherence trials (73.9% ± 9% vs. 95.7% ± 5.6%, main effect of coherence: $F_{(1,10)}$ = 165.107, p<0.001), while there was no significant effect of instruction (83.9% ± 8% during speed vs. 85.7% ± 7% during accuracy instructions, $F_{(1,10)}$ = 1.374, p=0.268) nor an interaction of instruction*coherence ($F_{(1,10)}$ = 1.152, p=0.308). To assess whether participants' behavior was, nevertheless, in line with our a-priori hypothesis that subjects would have decreased decision thresholds during speed instructions, we analyzed the RT distribution of error trials in the speed and accuracy conditions. Previous studies have shown that during conditions with high thresholds, errors are primarily observed during slow

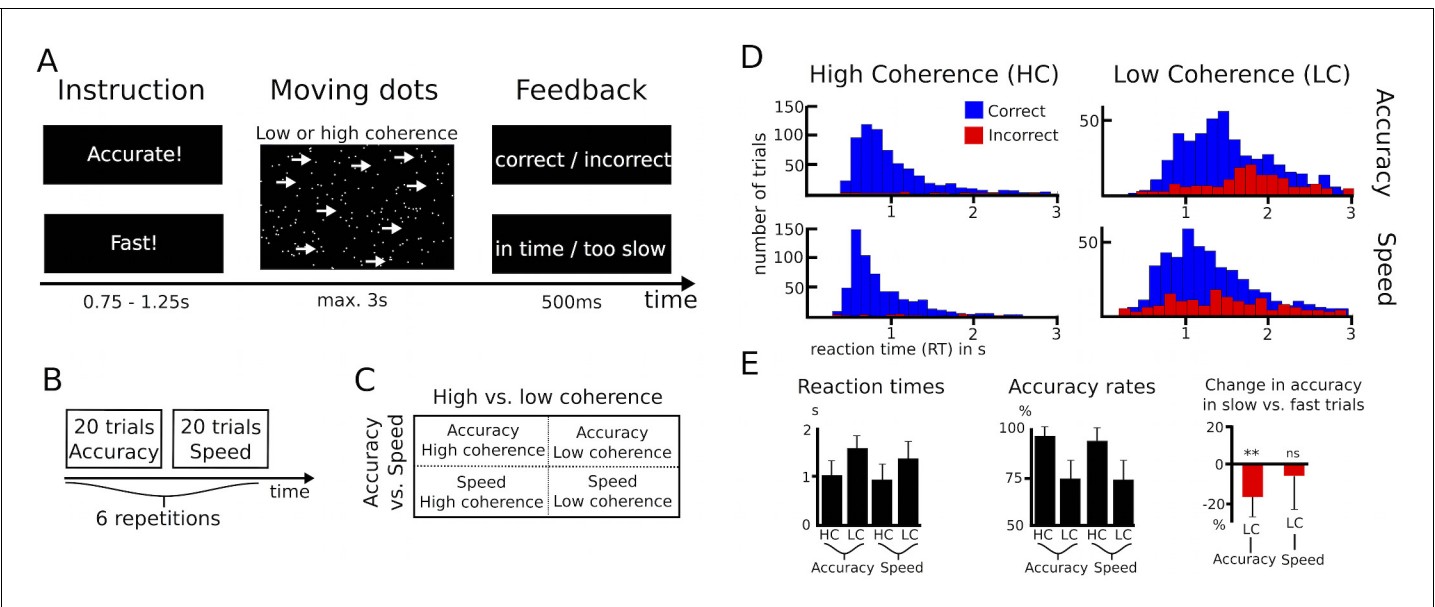

**Figure 1.** Paradigm and behavioral data. (A–C) Overview of experimental task and the 2*2 study design. (D) Histograms of RT distributions for correct (blue) and error (red) trials for all four conditions. (E) Second level comparison of reaction times (left), accuracy rates (middle) and change in accuracy rates in slow vs. fast trials (right). HC, high coherence; LC, low coherence; ** significant at p<0.01; ns, not significant. Error bars indicate standard deviation.

The following figure supplement is available for figure 1:

**Figure supplement 1.** Behavioral data of healthy participants.

responses (*Ratcliff and Rouder, 1998*; see Materials and methods for more details). In line with this, we found that participants made significantly more errors during slow compared to fast trials after accuracy instructions (16.6% ± 10.4% decrease in accuracy rates during slow responses, $t_{(10)} = -4.731$, $P_{corrected} = 0.002$), but not during speed instructions (6% ± 17.3%, $t_{(10)} = -1.641$, $P_{corrected} = 0.264$). Thus, even though there was no overall effect of speed instructions on accuracy rates, which is compatible with previous studies showing little effects on trials with very high or very low coherence (*Hanks et al., 2014*; *Ratcliff and McKoon, 2008*), our behavioral data were in line with the hypothesis that participants would decrease their decision thresholds during speed emphasis. To directly test this, we then applied the drift diffusion model (DDM), which allows computation of the latent decision-making parameters underlying participants' behavior.

## Drift diffusion model

DDM includes three main parameters of interest. First, the drift rate *v* reflects the rate of sensory evidence accumulation. Second, the decision threshold *a* determines the amount of sensory evidence that needs to be accumulated before the choice is executed. During speed-accuracy adjustments, when speed is required the decision threshold is thought to be decreased requiring less evidence before responding (*Ratcliff and McKoon, 2008*). Of note, an increased baseline level is mathematically equivalent to decreased decision thresholds (*Figure 2A*). Third, the non-decision time *t* reflects the portion of RT which is not directly related to the decision process, such as afferent delay, sensory processing and motor execution. In the current study, we assumed that drift rates were related to the coherence of the moving dots (low vs. high coherence) and thresholds were related to differences in task instructions (speed vs. accuracy). Furthermore, the non-decision time was allowed to vary between coherence and instruction conditions. We fitted this simple model to the data using a

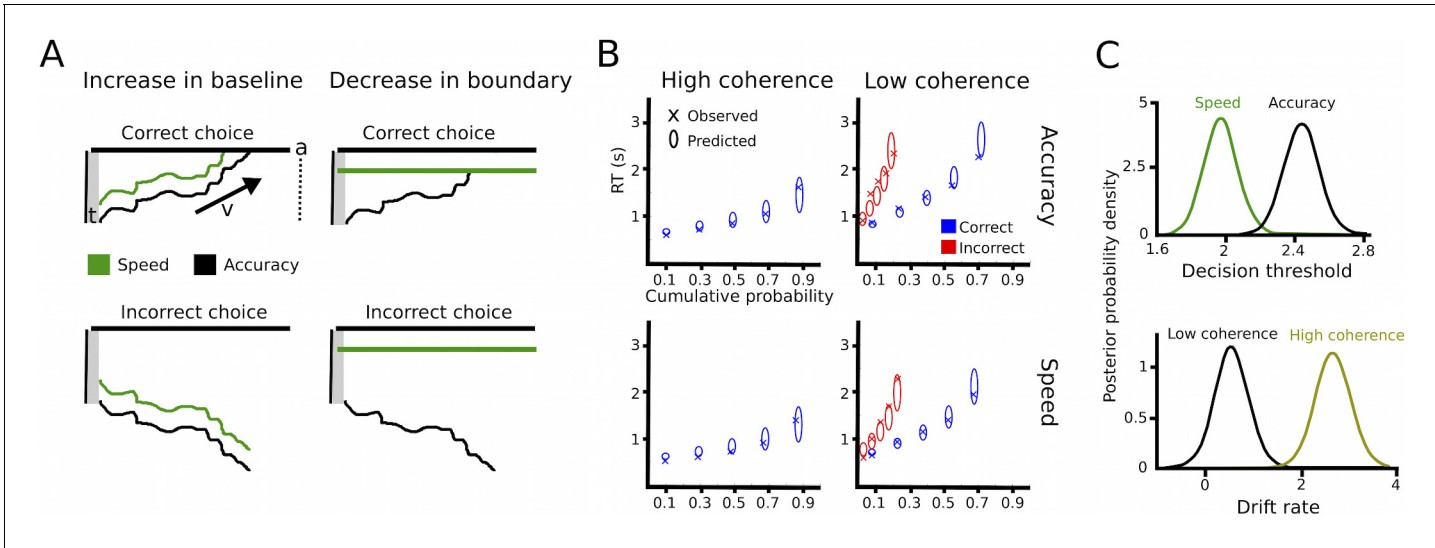

**Figure 2.** Drift diffusion modeling. (**A**) Schematic illustration of DDM. *t* is the non-decision time, *v* the drift rate and *a* the decision threshold. The upper boundary in the upper row indicates the threshold for the correct response, while the upper boundary in the lower row reflects the threshold for the incorrect response. Emphasizing speed over accuracy is thought to decrease the distance from the starting point of evidence accumulation to the decision threshold, which can be achieved by increasing the baseline (left column) or decreasing the boundary (right column). Both mechanisms are mathematically equivalent and cannot be distinguished in the DDM framework. (**B**) Quantile probability plots showing the observed (x) and predicted (ellipses) RT against their cumulative probabilities (10, 30, 50, 70 and 90 percentiles). The widths of the ellipses represent uncertainty (standard deviation of the posterior predictive distribution). Blue symbols are used for correct, red symbols for incorrect trials (incorrect trials are only shown for low coherence trials). (**C**) Posterior probability densities for changes in decision thresholds by instruction and changes in drift rates by coherence levels. Both effects were highly significant.

The following figure supplement is available for figure 2:

**Figure supplement 1.** Drift diffusion modeling in healthy participants.

hierarchical Bayesian estimation of DDM parameters (HDDM) and computed the posterior distribution of model parameters for statistical inference considering posterior probabilities ≥95% significant (*Wiecki et al., 2013*). The model fitted the data well, as indicated by accurate predictions of the observed RT distributions in all four conditions (*Figure 2B*). As expected, trials with low coherence had significantly lower drift rates than trials with high coherence (100% posterior probability) and decision thresholds after speed instructions were significantly lower than after accuracy instructions (>99% posterior probability, see *Figure 2C*). As a control analysis, we also assessed whether modulations of drift rates were related to changes in performance during speed vs. accuracy instructions, for example, due to increased attention to the stimuli. However, there was no effect of instruction on drift rates (67% posterior probability). Similarly, the non-decision time was neither modulated by instructions nor coherence (both 75% posterior probability). Thus, the HDDM analysis confirmed our a-priori hypotheses that changes in coherence of the moving dots would selectively alter drift rates, while speed vs. accuracy instructions would be related to adaptations of decision thresholds.

## Behavioral control experiment in healthy participants

To confirm that the observed behavior in Parkinson's disease patients resembled 'physiological' task performance, we additionally conducted the same task in 18 healthy age-matched participants (age of healthy participants: range 28–75 y, mean age 61 ± 16 y; age of Parkinson's disease patients: range 31–75 y, mean age 57 ± 12 y; difference between groups: $t_{(27)}$ = −0.675, p=0.505). In these healthy participants, RT were significantly faster in high compared to low coherence trials (652 ± 124 ms vs. 1238 ± 334 ms, main effect of coherence: $F_{(1,17)}$ = 65.218, p<0.001) and after speed compared to accuracy instructions (879 ± 194 ms vs. 1011 ± 219 ms, main effect of instruction: $F_{(1,17)}$ = 57.436, p<0.001, see *Figure 1—figure supplement 1*). There was also an interaction instruction*coherence ($F_{(1,17)}$ = 15.803, p=0.001), since RT decreased more strongly after speed instructions in low compared to high coherence trials (183 ± 121 ms vs. 81 ± 47 ms decrease in RT, $t_{(17)}$ = 3.924, p=0.001), which, however, did not remain significant when considering % change in RT ($t_{(17)}$ = 1.548, p=0.140). Importantly, neither the effect of coherence nor the effect of instruction on RT differed between patients and healthy controls when directly comparing the groups (effect of coherence $t_{(27)}$ = −0.793, p=0.435; effect of instruction $t_{(27)}$ = 0.809, p=0.425). Accuracy rates were lower in low compared to high coherence trials (98.9% ± 2.5% vs. 81.8% ± 6.6%, main effect of coherence: $F_{(1,17)}$ = 355.647, p<0.001), while there was no significant effect of instruction (89.9% ± 4.1% after speed vs. 90.9% ± 4.5% after accuracy instructions, main effect of instruction: $F_{(1,17)}$ = 2.193, p=0.157), nor an interaction instruction*coherence ($F_{(1,17)}$ = 0.599, p=0.450). Again there were no differences in the effect of coherence or instruction between patients and healthy participants (effect of coherence $t_{(27)}$ = −0.460, p=0.649; effect of instruction $t_{(27)}$ = 0.418, p=0.679).

We also tested whether there were differences in task-related changes in the latent decision-making parameters using HDDM (*Figure 2—figure supplement 1*). As in patients, we found that low coherence trials had significantly lower drift rates compared to high coherence trials (100% posterior probability). Speed instructions significantly reduced thresholds compared to accuracy instructions (>99% posterior probability), but had no significant effects on drift rates (79% posterior probability) or non-decision times (76% posterior probability). When comparing groups, we found that changes in coherence affected drift rates more strongly in healthy participants compared to patients (99% posterior probability). Importantly, however, there were no differences between groups regarding adjustments of decision thresholds (51% posterior probability), that is, patients and healthy participants changed their thresholds to a similar extent between speed and accuracy instructions.

Together, these findings show that task effects on performance and decision threshold adjustments were comparable between patients and healthy participants. In the next step, we aimed to elucidate the neural correlates of these effects by analyzing task-related changes in STN activity obtained from invasive LFP recordings in Parkinson's disease patients.

## Changes in STN activity during speed-accuracy adjustments

Aligning STN power to the onset of the motor responses showed the well-known decrease in 13–30 Hz beta power around the motor response (from ~250 ms before until 250 ms after the response), which was followed by an increase in beta power from ~500 ms until 1000 ms after the response

(*Figure 3A*). LFO power of 2–8 Hz showed an inverse pattern with a pronounced increase in power around the time of the response.

We found two neural correlates of speed-accuracy adjustments. First, when comparing speed and accuracy conditions, LFO power was significantly stronger after speed compared to accuracy instructions in the time period preceding the response (p<0.05), while there was no difference between trials with low and high coherence (p>0.05, see *Figure 3B–C*). The pre-response increase in LFO power during speed instructions was also observed when aligning the spectra to the onset of the moving dots cue with a significant cluster starting ~500 ms after dots onset (p<0.001), see *Figure 4A–C*. To assess whether the observed changes in LFO power were more strongly related to

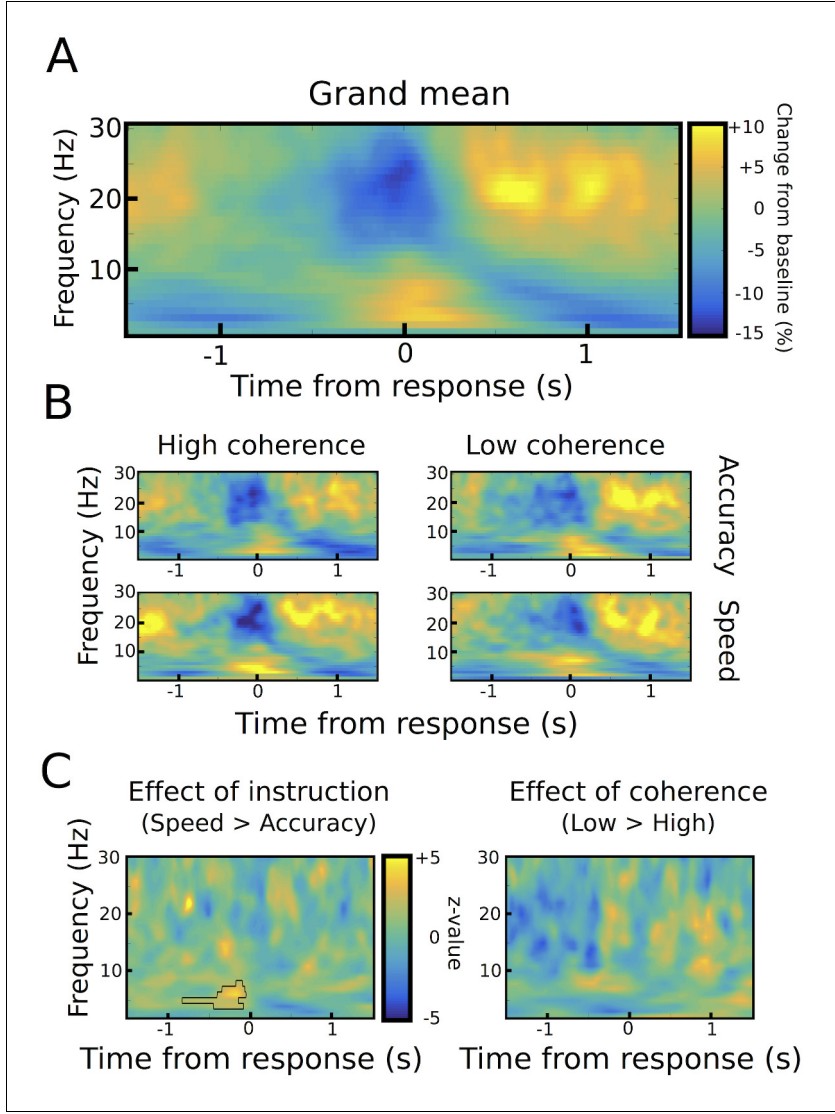

**Figure 3.** Response-aligned STN power changes. (**A**) Time frequency spectrum aligned to the onset of the motor response from −1.5 to +1.5 s averaged across conditions. (**B**) Spectra shown separately for the four conditions. The color map is identical to **A**. (**C**) Significant differences between conditions as revealed by cluster-based permutation tests.

The following figure supplements are available for figure 3:

**Figure supplement 1.** STN channel selection.

**Figure supplement 2.** Differences in STN power between correct and incorrect trials.

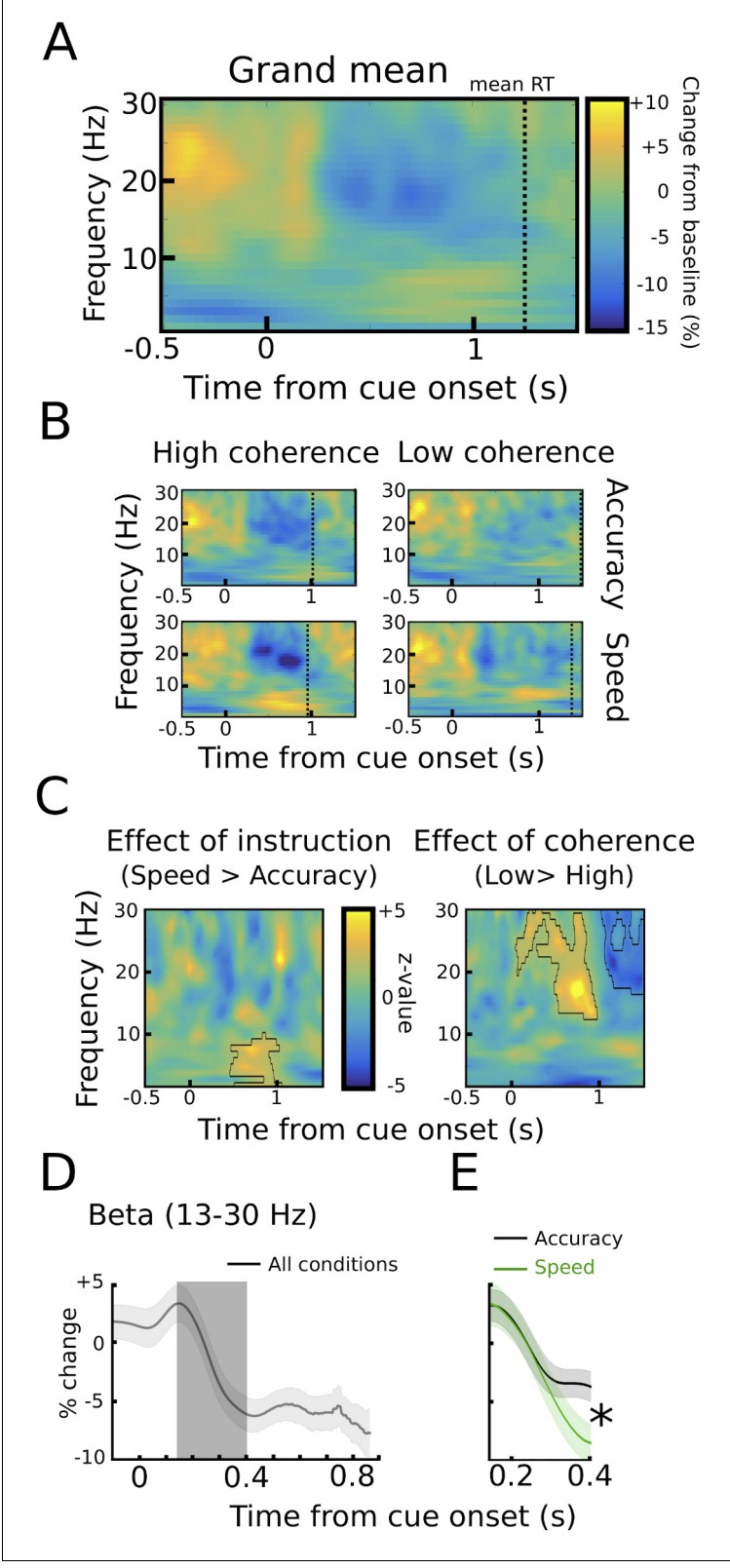

**Figure 4.** STN power changes aligned to the onset of the moving dots. (A) Time frequency spectrum aligned to the onset of the cue from −0.5 to +1.5 s averaged across conditions. (B) Spectra shown separately for the four conditions. The color map is identical to **A**. (C) Significant differences between conditions as revealed by cluster-based permutation tests. (D) Cue-induced decrease in beta power (13–30 Hz) averaged across conditions. The
*Figure 4 continued on next page*

*Figure 4 continued*

time series of each trial was capped at the time of the response before averaging. A decrease in beta power was evident from ~150 ms to 400 ms after the onset of the moving dots. (E) This beta decrease was stronger in speed vs. accuracy instructions (but not high vs. low coherence). For D and E and solid traces represent mean, while shaded areas around the traces indicate standard error of the mean. *, significant at p<0.05.

the onset of the moving dots cue or the response, we directly compared the mean LFO power during the respective time windows (from 750 ms before the response until the response vs. 500 ms until 1250 ms after the cue) after speed and accuracy instructions in a 2*2 ANOVA. Except from the already known main effect of instruction ($F_{(1,10)}$ = 10.941, p=0.008), this analysis did not show any significant differences between the cue- and response-related changes (effect of alignment: $F_{(1,10)}$ = 0, p=0.993, interaction alignment*instruction: $F_{(1,10)}$ = 0.93, p=0.358). Furthermore, in the considered time periods, there was no significant phase consistency of LFO across trials (intertrial-phase clustering, see Material and methods) in the cue- or response-aligned data neither after speed nor accuracy instructions (all p>0.05). This indicates that the observed differences in LFO power did not have strong cue- or response-evoked components.

We also found a correlate of speed-accuracy adjustments in the beta frequency band. For the cue-aligned spectra, there was a significant effect of coherence in the beta band with increased beta power from ~500 ms to 1000 ms after the cue and decreased beta power from ~1000 ms to 1500 ms after the cue in the low compared to high coherence trials (p<0.001), see *Figure 4C*. However, there was a very strong RT difference between the low and high coherence conditions (see dotted vertical lines in *Figure 4B*). Thus, the difference in the beta power might just reflect that responses took place later in low coherence trials. To address this effect of RT differences on STN beta power, we conducted an additional analysis based on (*Hanks et al., 2014*), in which we focused on the early cue-induced reduction in beta power (see Materials and methods for more details). This decrease was evident in all conditions (*Figure 4D*) and took place well before the second decrease in beta around the time of the response. ANOVA showed that the cue-induced beta decrease was more pronounced after speed compared to accuracy instructions (11.8% ± 5.8% vs. 6.9% ± 3.3%, main effect of instruction: $F_{(1,10)}$ = 6.619, p=0.028, *Figure 4E*), while there was no effect of coherence ($F_{(1,10)}$ = 0.177, p=0.683) or an interaction of instruction*coherence ($F_{(1,10)}$ = 2.525, p=0.143).

Together, we detected two correlates of speed-accuracy adjustments in the STN as reflected by task-related power changes; an enhanced cue-induced decrease in beta power from ~150 ms to 400 ms after onset of the moving dots followed by a stronger increase in LFO power starting ~750 ms before the response in speed compared to accuracy instructions. To further investigate these two features, we extracted single trial estimates of pre-response STN LFO power (from 750 ms before the response until the response) and the cue-induced beta power decrease (from 150 ms to 400 ms after the cue) and z-scored these values by subtracting the mean and dividing by the standard deviation for each subject (*Frank et al., 2015*; *Herz et al., 2016*), as in *Figure 5A*. Interestingly, these single trial changes in the LFO and beta band were not statistically related to each other as indicated by correlation analysis (mean rho = −0.006 ± 0.042, $t_{(10)}$ = −0.45, p=0.662, visualized in *Figure 5B*). Next, we aimed to test whether these changes were related to adjustments of decision thresholds as proposed by computational models of decision-making.

## Combining recorded STN power changes with drift diffusion modeling

Since we hypothesized that STN power changes related to speed-accuracy adjustments would modulate decision thresholds we entered single-trial estimates of STN LFO and beta power as predictors of decision thresholds into the HDDM regression analysis (see Materials and methods). This revealed a significant effect of LFO (99% posterior probability) as well as an interaction of LFO*instruction (95% posterior probability), see *Figure 5C*. Post-hoc tests showed that LFO predicted elevated decision thresholds only after accuracy instructions (96% posterior probability), but not after speed instructions (63% posterior probability). Thus, even though LFO was increased after speed instructions, it only predicted modulations of decision thresholds when subjects responded more cautiously after accuracy instructions (right column in *Figure 5C*). Cue-induced beta power decreases predicted lower decision thresholds (95% posterior probability), but there was no interaction of

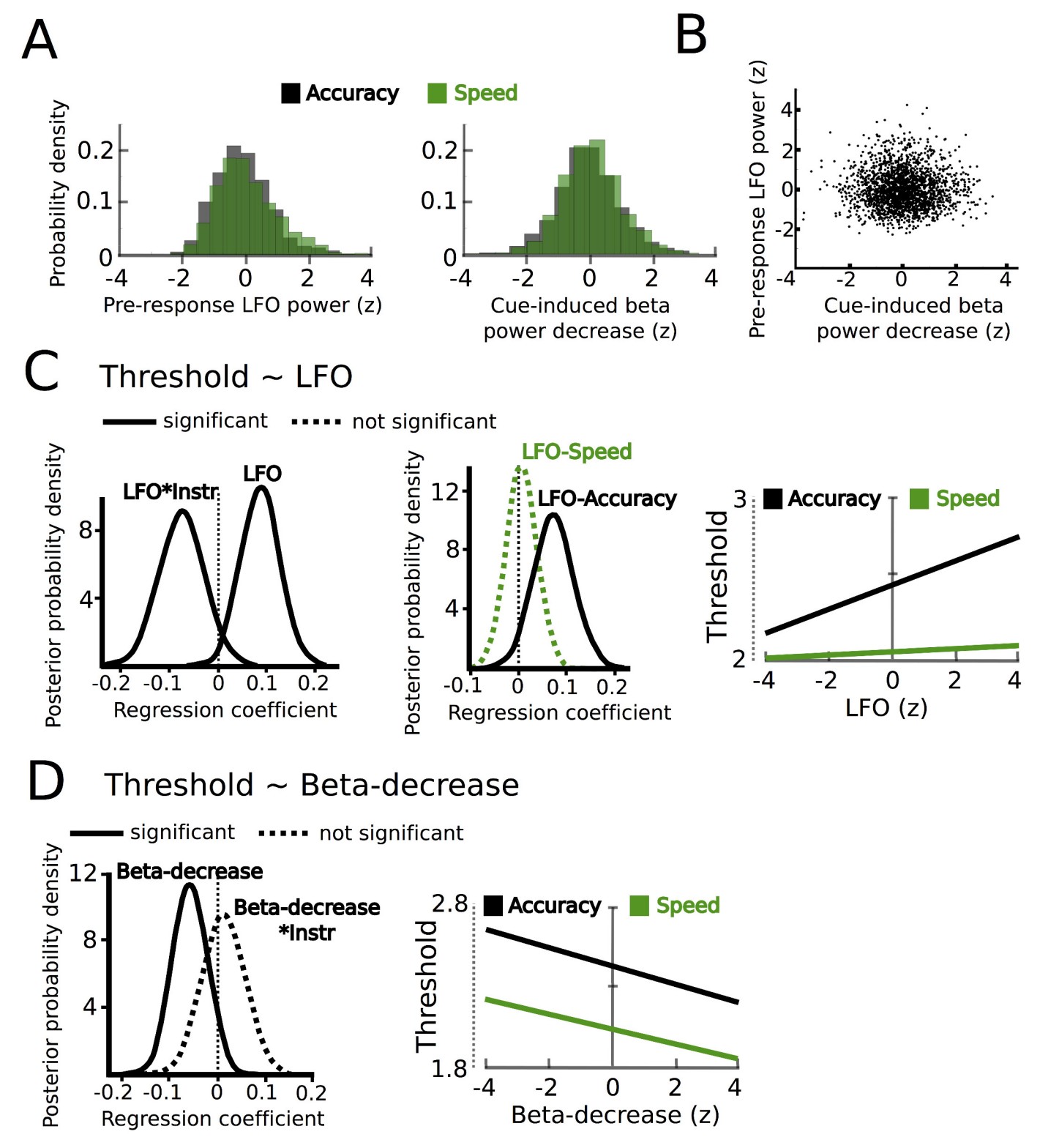

**Figure 5.** STN power changes predict adjustments of decision thresholds. (A) Histograms of the z-scored single trial values of the increase in LFO before the response (−750 ms before the response until the response) and cue-induced decrease in beta power (150 ms to 400 ms after the cue) for all subjects combined. Black bars represent trials with accuracy instructions, green bars trials with speed instructions. (B) Scatter plot of LFO and beta power single trial values illustrating the lack of a correlation between the two. Note that the statistical test for this analysis was based on a one-sample

*Figure 5 continued on next page*

*Figure 5 continued*

t-test of Fisher r-to-z-transformed within-subject correlation coefficients. (C) Posterior probability density for the effect of LFO on decision thresholds (left column), and the effect of LFO on thresholds separately for speed (green) and accuracy (black) instructions (middle column), which is further illustrated in the right column. (D) Posterior probability density for the effect of the cue-induced beta decrease on decision thresholds, which is further illustrated in the right column. Solid lines indicate significant results (≥95% of posterior distribution different from zero), while dotted lines indicate non-significant results.

beta\*instruction (59% posterior probability, *Figure 5D*). Thus, in contrast to LFO power changes, the relationship between beta power decreases and decision thresholds did not depend on speed vs. accuracy instructions. Importantly, the relationship between beta power decreases and thresholds was specific to the early cue-induced decrease in beta, since the averaged pre-response beta power (computed analogously to pre-response LFO power) did not predict threshold adjustments (54% posterior probability), which is in line with previous observations (*Herz et al., 2016*). As a further control analysis, we assessed whether STN power changes were specific to modulations of decision thresholds by conducting the same regression analysis using drift rates as dependent variable. This analysis showed that there were no relationships between STN LFO or beta power changes and trial-by-trial fluctuations in drift rates (effect of LFO: 69% posterior probability, LFO\*coherence: 73% posterior probability, beta-decrease: 53% posterior probability, beta-decrease\*coherence: 53% posterior probability). Finally, the HDDM incorporating STN power changes also improved model performance compared to the HDDM not containing any neural data (difference in deviance information criterion: 485).

To test whether the relationship between trial-wise changes in STN activity and decision thresholds were also reflected in simple behavioral measures, we conducted additional regression analyses using reaction times as dependent variable and STN activity (LFO and beta power, see above) as predictors. We found that the pre-response increase in LFO power predicted increased RT only after accuracy (slope: 0.038, p=0.022), but not after speed instructions (slope: −0.006, p=0.366). In addition, stronger cue-induced decreases in beta power predicted shorter RT across instructions (slope: −0.022, p=0.035). These findings show that similar effects of STN activity can be detected using either DDM or simple behavioral measures. However, applying DDM allowed us to relate STN activity changes to a specific latent decision-making parameter, that is, the decision threshold. This is not possible through correlations with behavioral measures alone, since RT is not only influenced by changes in decision thresholds, but also the rate of evidence accumulation and processes not directly related to the decision process (including motor execution delays).

Together, combining recordings of STN power changes during speed-accuracy adjustments with HDDM demonstrated that the observed changes in the LFO and beta band both significantly contributed to trial-by-trial modulations of decision thresholds. Since STN oscillations with different spectral properties have been related to distinct oscillatory cortico-STN networks (*Litvak et al., 2011*), we aimed to investigate whether the observed changes in STN LFO and beta power were related to modulations of different cortico-STN connections in the final part of the analysis.

## Modulation of cortico-STN connectivity during speed-accuracy adjustments

While cortical oscillatory activity in the beta band is a hallmark of motor networks (*Litvak et al., 2011*; *Pfurtscheller, 1981*; *2003*), cortical LFO activity during 'cognitive' motor tasks has mainly been observed in PFC (*Cavanagh et al., 2011*, *2012*; *Cohen, 2014b*). Therefore, we assessed whether task-related modulations of the beta and LFO rhythms were predominantly expressed at scalp electrode Fz (over PFC) vs. scalp electrodes C3/C4 (over or close to motor cortex) as a first step (*Figure 6A and B*). EEG activity recorded at Fz showed a stronger pre-response LFO increase compared to electrodes C3/C4 (2.7% ± 4.4% vs. −1.1 ± 1.3%, p<0.05). Conversely, activity at C3/C4 displayed a more pronounced cue-induced decrease in beta power compared to Fz (12% ± 8% vs. 4.3% ± 1.8%, p<0.001).

To directly assess whether oscillations in the beta and LFO band observed in the STN might be related to changes in cortico-STN connectivity, we then computed the inter-site phase clustering (ISPC), a phase-based connectivity measure, which indicates how reliably the phases of oscillatory

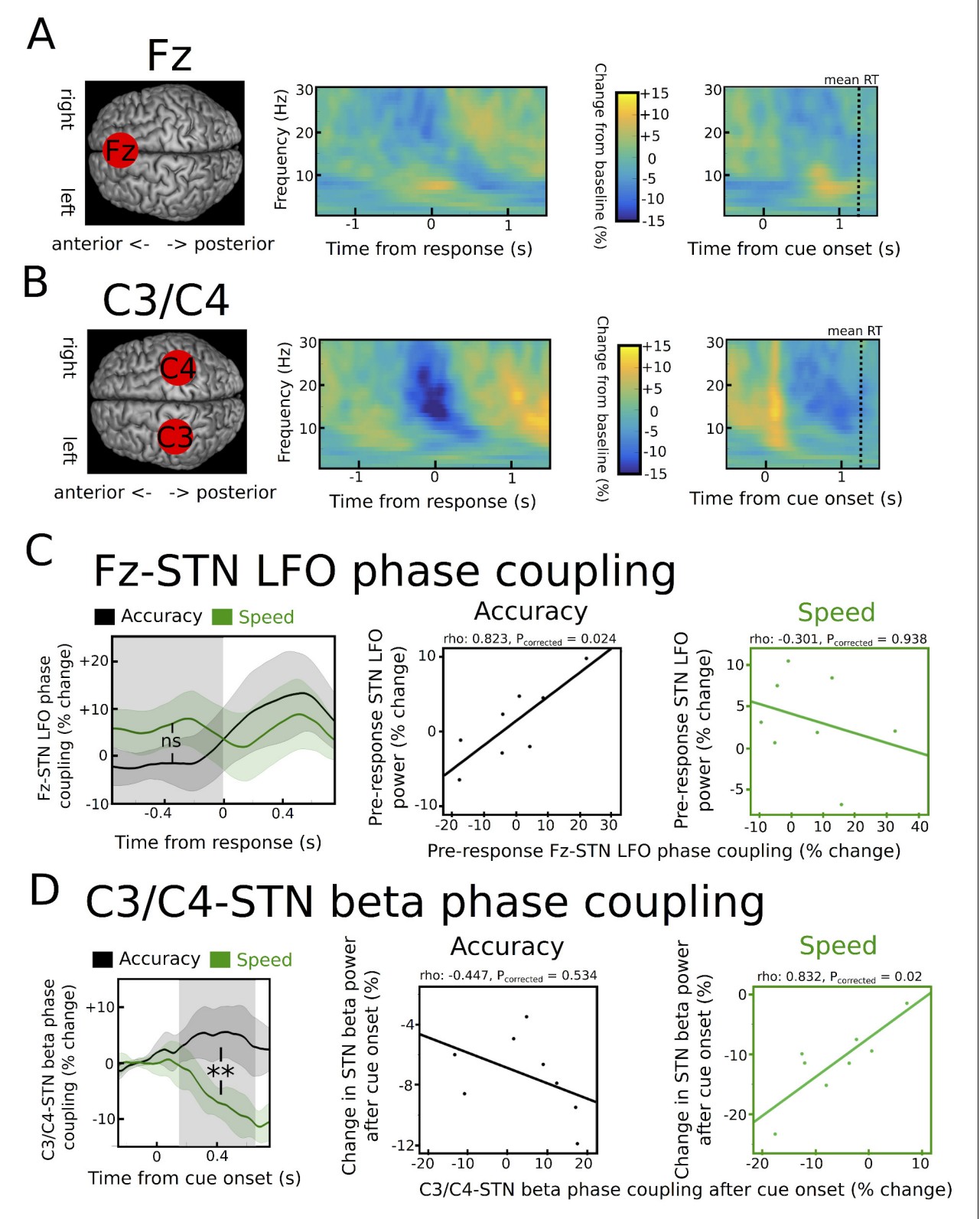

**Figure 6.** Task-related changes in cortical activity and cortico-STN connectivity. (**A**) Time frequency spectrum for EEG at electrode Fz aligned to the motor response (from −1.5 to +1.5 s) and cue-onset (from −0.5 to 1.5 s) averaged across conditions. The pre-response increase in LFO was significantly stronger at Fz compared to electrode C3/C4 (p<0.05). (**B**) Time frequency spectrum for EEG at electrode C3/C4 aligned to the motor response (from −1.5 to +1.5 s) and cue-onset (from −0.5 to 1.5 s) averaged across conditions. The cue-induced beta decrease was significantly stronger at C3/C4

*Figure 6 continued on next page*

*Figure 6 continued*

compared to Fz (p<0.001). (C) Inter-site phase clustering (ISPC) between LFO at Fz and STN. The left column shows ISPC separately for speed (green) and accuracy instructions (black). In C and D, the shaded rectangles indicate the time windows of interest, which were based on the analysis of STN power changes. Solid traces represent mean and shaded areas around the traces indicate standard error of the mean. The middle and right columns show correlation analyses between Fz-STN LFO ISPC and STN LFO power changes for accuracy and speed instructions, respectively. Each observation corresponds to one patient (n = 8). (D) Same as C, but for C3/C4-STN ISPC in the beta band. **, significant at p<0.01; ns, not significant.

activity in two areas are aligned across trials (*Cohen, 2014a*; *Herz et al., 2016*; *Zavala et al., 2014*). We assessed whether cortico-STN phase coupling was different after speed vs. accuracy instructions and whether inter-individual differences in cortico-STN connectivity were correlated with differences in STN power during speed and accuracy instructions. The latter analysis was important in order to test if the context-specific relationship between STN-LFO power and decision thresholds, which was only observed after accuracy instructions, might be related to an increased influence of the PFC underlying Fz on the STN during modulation of decision thresholds when participants are more cautious (i.e. after accuracy instructions). This is predicted by neurobiological models of decision-making (*Frank, 2006*; *Ridderinkhof et al., 2011*) and suggested by previous studies (*Frank et al., 2015*; *Herz et al., 2016*).

Fz-STN ISPC in the LFO band was not significantly different between speed and accuracy instructions in the pre-response period (p>0.1), see *Figure 6C*. However, correlations between Fz-STN ISPC and STN LFO power were significantly different between speed and accuracy instructions ($z_{(7)}$ = −2.33, $P_{corrected}$ = 0.04). While there was no relationship between Fz-STN ISPC and STN power after speed instructions (rho = −0.301, $P_{corrected}$ = 0.938), there was a significant positive correlation between Fz-STN coupling and STN power in the LFO band after accuracy instructions (rho = 0.823, $P_{corrected}$ = 0.024), see *Figure 6C*. Thus, only after accuracy instructions, in which single-trial STN LFO power predicted adjustments of decision thresholds, were inter-individual differences in STN LFO power closely related to differences in Fz-STN phase coupling. This relationship between Fz-STN LFO phase coupling and STN LFO power was absent after speed instructions, similar to the lack of a relationship between single-trial STN LFO power and trial-wise modulations of decision thresholds when speed was emphasized.

C3/C4-STN ISPC in the beta band showed a pronounced decrease after onset of the cue only after speed, but not accuracy instructions (*Figure 6D*), leading to a significant difference between conditions (p<0.01). Importantly, this difference is unlikely to be a simple consequence of the stronger cue-induced STN beta power decrease after speed vs. accuracy instructions, since analyzing ISPC 300 ms after the onset of moving dots, in which there was no difference between instructions in STN beta power (see *Figure 4E*), still yielded a significant difference in C3/C4-STN ISPC (p<0.05). Correlation analyses revealed that the relationship between C3/C4-STN beta ISPC and STN beta power was significantly different between speed and accuracy instructions ($z_{(7)}$ = −2.65, $P_{corrected}$ = 0.016). This was driven by a significant positive correlation between C3/C4-STN ISPC and STN beta power after speed instructions (rho = 0.832, $P_{corrected}$ = 0.02), while there was no significant correlation after accuracy instructions (rho = −0.447, $P_{corrected}$ = 0.534; *Figure 6D*) when STN may have been hijacked by coupling with Fz.

## Discussion

In the current study, we combined recordings of STN LFP and cortical EEG during perceptual decision-making with computational modeling of the latent parameters underlying the decision process. This allowed us to demonstrate neural correlates of the involvement of STN and of cortico-STN connectivity in controlling the trade-off between fast and accurate decisions. We found two distinct, statistically-uncorrelated mechanisms comprising modulations of a Fz-STN network in the LFO band and a C3/C4-STN network in the beta band. These two networks differed in their cortical topography, relative timing of recruitment and spectral characteristics as well as in their precise relationship with decision thresholds. The Fz-STN network predicted increased thresholds only after accuracy instructions, while the C3/C4-STN network predicted decreased thresholds irrespective of instructions, but was more strongly modulated during speed emphasis.

Electrophysiological recordings of STN LFPs and EEG have shown that LFO modulations during 'cognitive' motor tasks are mainly related to cortical PFC activity and PFC-STN connectivity (*Cavanagh et al., 2011*, *2012*; *Cohen, 2014b*; *Herz et al., 2016*; *Pastötter et al., 2012*; *Zavala et al., 2014*). More specifically, *Pastötter et al. (2012)* demonstrated that changes in cortical LFO power are mainly localized in the medial PFC using EEG in healthy participants performing a speed-accuracy trade-off task. Furthermore, *Frank et al. (2015)* showed that trial-by-trial variations in Fz LFO measured using EEG correlated with activity of the medial PFC measured using fMRI and that both signals were related to increases in decision thresholds. These findings add to the increasing evidence that the PFC contributes to decision-making and modulations of the speed-accuracy trade-off (*Aron et al., 2016*; *Bogacz et al., 2010*; *Frank, 2006*; *Heekeren et al., 2008*). Within the PFC, mainly – albeit not exclusively - dorsomedial PFC seems to play a pivotal role during context-dependent adaptations of decision-making and motor control (*Forstmann et al., 2010*, *2008*; *Frank et al., 2015*; *Herz et al., 2014*; *Ivanoff et al., 2008*; *van Veen et al., 2008*; *Wenzlaff et al., 2011*). Importantly, dorsomedial PFC has anatomical connections to the STN via the indirect and hyperdirect pathway (*Alexander et al., 1986*; *Aron et al., 2007*; *Forstmann et al., 2010*; *Lambert et al., 2012*). Here, we found that STN LFO power was increased prior to the response after speed compared to accuracy instructions. However, it was only predictive of elevated thresholds after accuracy instructions and, similarly, the extent to which Fz and STN LFO fell into register (as reflected by phase coupling) only correlated with variations in STN LFO power after accuracy instructions. These observations are in line with the idea that PFC increases its influence over STN activity when caution is warranted (*Frank, 2006*; *Ridderinkhof et al., 2011*) leading to increased decision thresholds in order to delay the response (*Frank et al., 2015*; *Herz et al., 2016*). Further evidence for this hypothesis has been provided by previous studies assessing 'conflict' between competing responses. Using the Flanker task Zavala and colleagues (2013, 2015, 2016) found increased STN LFO power in trials with distracting stimuli compared to trials without such conflict. Similar changes in STN LFO power have been observed during a Stroop task (*Brittain et al., 2012*). Since response conflict has been shown to increase decision thresholds (*Ratcliff and Frank, 2012*), these previously reported LFO changes might be related to threshold adjustments. However, LFO activity – at least when measured over PFC using EEG - cannot be simply mapped to one specific process or mechanism, since it is not only modulated by conflict, but various processes including novelty, punishment, error, memory and learning (*Cavanagh et al., 2012*; *Cohen, 2014b*). Thus, the increased STN LFO power after speed instructions observed in the current study could be related to a variety of processes, such as the implementation of a non-default mode of responding (*Fleming et al., 2010*) assuming that participants' natural tendency was to weigh accuracy over speed (*Forstmann et al., 2008*). Importantly, however, this change in LFO power after speed emphasis was not related to adjustments of decision thresholds nor to PFC-STN phase coupling indicating either that cortical areas other than dorsomedial PFC were increasing their influence over STN or that communication between PFC and BG were mainly relayed through the striatum instead of STN for threshold adjustments during the speed regime (*Forstmann et al., 2008*; *Ivanoff et al., 2008*; *van Veen et al., 2008*).

We found a separate neural correlate of speed-accuracy adjustments, namely an immediate cue-induced decrease in STN beta power, which was more pronounced during speed emphasis and predicted lower decision thresholds irrespective of the type of instruction. Furthermore, variations in STN beta power after speed instructions were closely related to the extent to which beta phase coupling between C3/C4 and STN decreased after the cue indicating that the stronger STN beta power decrease during speed emphasis was related to modulations of C3/C4-STN connectivity. It should be noted that concomitant changes in power can affect measures of phase coupling, since low power can render phase estimates unreliable underestimating the consistency of phase-alignment between areas. Thus, the decreased phase coupling after speed instructions should be interpreted with caution, even though we found that C3/C4-STN beta phase coupling was already decreased when STN beta power showed no differences between speed and accuracy instructions ~300 ms after the cue. At the cortical level, beta power modulations during movements (*Pape and Siegel, 2016*; *Pfurtscheller, 1981*; *2003*), decision-making (*Donner et al., 2009*; *Wyart et al., 2012*) and speed-accuracy trade-offs (*Pastötter et al., 2012*) have been localized to motor areas. Furthermore, STN beta oscillations are coherent to motor cortical areas during rest (*Hirschmann et al., 2011*; *Litvak et al., 2011*). Thus, even though conclusive evidence can only be provided through

simultaneous STN and focal cortical recordings, for example, using electrocorticography, the observed beta band changes in STN and at C3/C4 in the current study are likely to be related to modulations of a motor cortical-STN network adding to the increasing evidence that 'decision-related' signals can be observed in areas involved in motor preparation and execution (*Cisek and Kalaska, 2010*; *Donner et al., 2009*; *Klein-Flügge and Bestmann, 2012*; *Pastötter et al., 2012*; *Thura and Cisek, 2016*). One role of motor cortical-STN beta modulations during speed-accuracy adjustments might be related to the amount of vigor invested in the response. In the current study, we did not measure movement velocity or force. However, previous studies have provided strong evidence that the BG in general (*Desmurget and Turner, 2010*; *Yttri and Dudman, 2016*), and variations in STN beta power in particular (*Tan et al., 2015*, *2016*), are related to encoding movement vigor. During speed-accuracy adjustments speed emphasis not only decreases decision times, but also increases movement vigor to indicate the choice (*Spieser et al., 2016*; *Thura et al., 2014*). Furthermore, the longer is spent on deliberation the less time is spent on performing the motor response (*Thura and Cisek, 2016*; *Thura et al., 2014*). Such close relationships between 'cognitive' and 'motor' aspects of a decision have led to the hypothesis that a common signal might underlie adjustments in the speed of decision and vigor of movements (*Thura and Cisek, 2016*; *Thura et al., 2014*). Future studies are needed to test the intriguing possibility that modulations of cortico-STN networks during speed-accuracy adjustments do not only mediate changes in decision time, but also related changes in movement vigor.

How can changes in cortico-STN connectivity lead to adjustments between fast and accurate decisions? Recordings of cortical single-unit activity in non-human primates have shown that when speed is emphasized, neurons integrating sensory evidence exhibit an increase in the baseline and gain of firing rates during deliberation, rather than terminating at lower levels of firing rates before the choice is executed (*Hanks et al., 2014*; *Heitz and Schall, 2012*; *Thura and Cisek, 2016*). These findings indicate that a shift in baseline, but not a change in the threshold (which is mathematically equivalent, see also *Figure 2A*), underlies speed-accuracy adjustments at the neural level. Such changes in cortical activity and excitability could be mediated through the BG by modulating tonic feedback inhibition of the cortex (*Alexander et al., 1986*; *Bogacz et al., 2010*; *Kravitz et al., 2010*) putatively in the form of a dynamic 'urgency' signal (*Churchland et al., 2008*; *Cisek and Kalaska, 2010*; *Drugowitsch et al., 2012*; *Thura and Cisek, 2016*; *Thura et al., 2014*). Interestingly, changes in cortical firing rates during speed-accuracy adjustments have been observed in multiple areas comprising motor and premotor cortex (*Thura and Cisek, 2016*), parietal cortex (*Hanks et al., 2014*) and frontal eye field (*Heitz and Schall, 2012*), all of which share connections with the STN (*Lambert et al., 2012*). Thus, context-dependent changes in STN activity could adjust cortical activity (e.g. baseline firing rates) simultaneously in multiple, spatially remote areas according to current task demands. As mentioned earlier, cortex and basal ganglia are interconnected through circuits, which comprise not only cortico–STN connections, but involve a multitude of subcortical areas (*Alexander et al., 1986*; *Kravitz et al., 2010*; *Lambert et al., 2012*). Therefore, we do not assume that during speed-accuracy trade-offs neural activity changes within the basal ganglia are limited to the STN. Behavioural adjustments might ultimately be implemented by modulations of cortico-basal ganglia network dynamics through multiple pathways. Since spike-based communication can be optimized regarding efficiency and selectivity through synchronized oscillations (*Fries, 2015*), we propose that the changes in oscillatory activity observed in this study reflect modulations of distinct neural communication channels facilitating adjustments between hasty and cautious decision regimes.

Together, our results suggest that distinct mechanisms are employed in the brain to facilitate speed-accuracy adjustments in cortico-basal ganglia networks. A better understanding of such mechanisms might render it possible to focus therapeutic interventions on specific neural circuits in order to improve treatment of neurological disorders in the future.

## Material and methods

### Participants

Eleven patients with Parkinson's disease (PD, ten males, mean disease duration 7.8 years ± 2.9 standard deviation (SD), mean age 56.7 years ± 11.7 SD), who had undergone deep brain stimulation

(DBS) surgery of the bilateral subthalamic nucleus (STN) 2–5 days prior to the recordings, were enrolled in the study. For more clinical details of the patients and specifications of the inserted electrodes please see *supplementary file 1*. Lead localization was verified by stereotactic intraoperative magnetic resonance imaging or by monitoring the clinical effect and side effects during operation and immediate postoperative stereotactic computerized topography. Recordings from bilateral STN were performed through externalized electrode extension cables in the time period between electrode insertion and implantation of the subcutaneous pacemaker approximately 1 week after the first operation.

In order to approximate physiological STN function as well as possible all patients were tested on their normal dopaminergic medication. All patients had a good dopamine response as indicated by a pronounced improvement of motor function after a levodopa challenge in the preoperative assessment (mean improvement in Unified Parkinson's Disease Rating Scale-III 61.9% ± 12.6, see *supplementary file 1*). Of note, decision thresholds - the main interest of the current study – are not modulated by dopamine in healthy people (*Winkel et al., 2012*) or PD patients (*Huang et al., 2015*). To assess whether patients' behavior was comparable to that of healthy controls, we also recruited 18 healthy age-matched control participants (eight males, mean age 60.5 ± 16.1 years). In accordance with the declaration of Helsinki, participants gave written informed consent to participate in the study, which was approved by the local ethics committee (Oxfordshire REC A).

## Experimental task

We used a modified version of the moving dots task; see *Figure 1A*. The task was presented on a MacBook Pro (OS X Yosemite, version 10.10.3, 13.3 inch Retina display, 60 Hz refresh rate) using PsychoPy v1.8 (*Peirce, 2007*; RRID:SCR_006571). The display was viewed from a comfortable distance allowing the subjects to interact with the keyboard. At the beginning of each trial, a text cue indicated whether participants should respond as quickly ('Fast!') or as accurately as possible ('Accurate!'). The duration of this cue was randomly jittered between 0.75 and 1.25 s with an average duration of 1 s. Then, a cloud of 200 randomly moving white dots was presented on a black background. The diameter of the cloud was 14 cm and dot size was 10 pixels. Each dot moved in a straight line at a rate of 0.14 mm per frame for 20 frames before moving to another part of the cloud where it moved in a new direction chosen pseudorandomly between −180° and 180°. While some of the dots were moving randomly, the remaining dots moved coherently in one direction, which made the cloud of dots appear to move to the left or right. Participants were instructed to press a key with their right index finger ('/' on the keyboard of the laptop) if they perceived that the cloud was moving to the right and to press a key with their left index finger ('z') when they perceived a leftwards movement. Between responses both index fingers rested on the respective keys. The percentage of dots moving coherently in one direction was either 50% (high coherence) or 8% (low coherence). These two cues were pseudorandomly presented with equal probability so that participants could not predict whether the next trial would contain dots with high or low coherence. The trial was terminated by a response or after a 3 s deadline in case participants did not respond followed by immediate visual feedback, which was shown for 500 ms. During accuracy instruction 'incorrect' was shown as feedback both for errors of commission and errors of omission, while 'correct' was shown for all correct trials. During speed instructions 'in time' was shown for all responses within the 3 s window, while 'too slow' was shown if patients did not respond within the 3 s deadline. Similar to previous studies of speed-accuracy adjustments in PD, we did not impose a more restricted time window for responding during speed instruction (*Green et al., 2013*; *Huang et al., 2015*), since motor function varies considerably between PD patients. While trials with different coherence levels were randomly interspersed, accuracy and speed trials alternated in blocks of 20 trials (*Green et al., 2013*). These blocks were repeated six times each resulting in 240 trials for the whole experiment (*Figure 1B*), which lasted approximately 10 min. Before commencement of the experimental recordings, patients could practice the task for as long as they wished (usually approximately 40 trials).

## Analysis of behavioral data

Prior to statistical analyses, trials without responses (errors of omission) or reaction times (RT) <0.25 s were excluded (combined 3.1% of all trials). The trials differed regarding the type of instruction (accuracy vs. speed) and coherence (high vs. low); see *Figure 1C*. Accordingly, RT were analyzed

using a 2*2 ANOVA using SPSS statistics v22 (IBM, New York, USA; RRID:SCR_002865). In case of significant interactions, post-hoc tests were conducted using paired samples t-tests. Since the RT distribution of error trials is important for inferences on latent decision-making parameters in the drift diffusion model (DDM) framework (see below), we additionally analyzed whether errors were primarily observed during fast or slow responses. To this end, we divided erroneous responses during low coherence trials (there were only ~5% errors in high coherence trials, see Results) into fast and slow trials after a median split. We then calculated the % change in accuracy by computing ($Accuracy_{slow}$ – $Accuracy_{fast}$) / $Accuracy_{fast}$ separately for Accuracy and Speed instructions and tested whether the resulting values were different from 0 using one-sample t-tests. To directly compare patients and healthy participants, we computed the effect of instruction (accuracy vs. speed) and coherence (low vs. high) on RT and accuracy and calculated independent samples t-tests. For all tests, the statistical threshold was set to p=0.05. Bonferroni correction was applied when appropriate and corresponding p-values marked as $P_{corrected}$ for clarity. Normality assumptions were tested using Shapiro-Wilk tests. Since accuracy rates violated the normality assumption, we applied an arc-sine transform (inverse of the sine function) before statistical testing, but report non-transformed % values in figures and main text for readability. Values for statistical analyses are given as mean ± standard deviation throughout the article unless stated otherwise.

## Processing of electrophysiological data

We recorded bilateral STN LFPs from the implanted DBS electrodes. Additionally, in 8 of the 11 patients, electroencephalography (EEG) was recorded over two regions of interest: dorsomedial prefrontal cortex (PFC) and motor cortex areas. These regions were recorded from Fz as well as C3 (left hemisphere) and C4 (right hemisphere) according to the international 10–20 system. We also recorded from midline electrodes Cz, Pz and Oz whenever possible, but due to surgical wounds and dressings we were not able to cover a broader area. Accordingly, although we maintain that Fz, C3 and C4 recorded activity from our regions of interest, we do not assume that they do so exclusively. Electrooculography (EOG) was recorded for eye movement artifact rejection. Data were sampled at 2048 Hz, band-pass filtered between 0.5 and 500 Hz and amplified (TMSi porti, TMS International, Enschede, The Netherlands). Further analyses were conducted offline using custom-written Matlab scripts (R2015a, The MathWorks, Natick, MA, USA; RRID:SCR_001622). Artifacts related to eye movements were removed by subtracting the filtered and adaptively scaled EOG data (40 Hz low-pass filter). Scaling was performed with an optimization algorithm (Matlab function *fminocn*, initial value = 1) minimizing the sum of squared errors (*Fischer et al., 2016*). Trials with residual artifacts were discarded after visual inspection. After removal of trials based on behavioral data (see above) and artifacts in the electrophysiological data on average 201 trials (83.4%) remained per patient resulting in 2209 trials combined. Data were downsampled to 200 Hz, high-pass filtered at 1 Hz and low-pass filtered at 100 Hz. For quadripolar contacts (five patients, see *supplementary file 1*), LFPs were converted to a bipolar montage between adjacent contacts (three channels per hemisphere) to limit effects of volume conduction (*Herz et al., 2016*). Similarly, for octopolar non-directional contacts (three patients), bipolar channels were computed between adjacent contacts leading to seven channels per hemisphere. For octopolar directional contacts (three patients), bipolar montages were created between the dorsal omnidirectional contact and its three adjacent directional contacts as well as between the ventral omnidirectional contact and its three adjacent directional contacts resulting in six bipolar channels per hemisphere. Power and phase of LFPs were computed using the continuous wavelet transform with two cycles per frequency for the lowest considered frequency (2 Hz) which linearly increased to five cycles per frequency for the highest considered frequency (30 Hz) in 1 Hz steps. We chose this frequency range, because we *a-priori* expected low-frequency oscillations (LFO) between 2 and 8 Hz and beta oscillations between 13 and 30 Hz to be modulated during the task (*Herz et al., 2016*; *Zavala et al., 2014*). Power of each frequency was normalized to the mean signal of that frequency across the whole experiment. Of note, we applied normalization to account for between-subject differences unrelated to the task (e.g. signal-to-noise ratio), however, observed qualitatively highly similar spectra when omitting this step. The resulting time frequency spectra were aligned to the onset of the moving dots and time of the response, respectively. In order to detect the bipolar STN channel, which was most strongly modulated by the task, we employed the following procedure. First, for each patient and hemisphere, response-aligned time-frequency spectra (averaged across all conditions to avoid circularity) were visualized for all channels and the

channel with the most pronounced beta modulation (reduction of beta power in the peri-response time window) was noted. For further analyses, we then selected the channel showing the strongest peri-response LFO increase if this channel (i) also showed a strong beta-modulation (>15% change from baseline) and (ii) was not further than two channels remote from the best beta channel to avoid including activity likely recorded outside of the STN (contact length is 1.5 mm with 0.5 mm spacing between contacts for all implanted electrodes). In 53% of hemispheres, the bipolar channel showing the strongest beta modulation was identical to the channel showing the strongest LFO modulation and in 47% the best LFO channel was localized ventrally to the best beta channel. *Figure 3—figure supplement 1* shows an example of STN channel selection for one representative patient. When using the best beta channel instead of the best LFO channel for analyses of task-related modulations of STN activity, we obtained highly similar results regarding modulations in the beta band to the ones presented in the paper (data not shown). After channel selection, normalized spectra of the selected left and right STN were averaged resulting in one STN channel per patient.

Preprocessing and time-frequency transformation of EEG channels were identical to the procedure for STN LFPs described above except that EEG channels were not converted to a bipolar montage, but referenced to the average of all EEG channels connected to the TMSi porti during recordings (www.tmsi.com). Analysis of C3 activity was performed for right hand movements and of that from C4 for left hand movements, while for analysis of PFC activity, the midline electrode Fz was used. Of note, since EEG activity in prefrontal areas often has a clearer lower boundary at ~4 Hz - in contrast to STN activity, which typically extends into lower frequencies (*Cavanagh et al., 2011*; *Herz et al., 2016*; *Zavala et al., 2014*) - it is often referred to as theta power (*Cavanagh et al., 2011*, *2012*; *Cohen, 2014*). Here, however, we use the same 2–8 Hz band for both STN and EEG power for consistency, and term this LFO activity.

## Statistical analysis of STN LFPs

For the statistical analysis, we compared time-frequency spectra during speed vs. accuracy instructions (effect of instruction) and low vs. high coherence trials (effect of coherence) using cluster-based permutation testing (*Maris and Oostenveld, 2007*) by shuffling between condition labels for each subject. We applied 1000 permutations and used p=0.05 as cluster-building threshold and as statistical threshold for cluster-based comparisons. The time window of interest was 1.5 s prior to until 1.5 s after the movement for the response aligned spectra and 0.5 s prior to until 1.5 s after the onset of moving dots for the cue-aligned spectra. We also compared differences between error and correct trials for the low coherence condition. This control analysis showed no significant differences and is shown in *Figure 3—figure supplement 2*.

In order to take into account RT differences between conditions in the cue-aligned spectra we applied an additional analysis based on (*Hanks et al., 2014*) for changes in the beta band. This was necessary, because the well-known strong beta-decrease during movements will necessarily induce strong differences in cue-aligned beta power changes between conditions with short and long RT, simply because the response will take place earlier in the condition with short RT. Thus, for each trial, we computed the change in beta power over time until the response was executed, that is, the time series of each trial was capped at the time of the response. We then averaged these time series across trials from 500 ms before dots onset until the point in time when 50% of trials in the condition with lowest RT (high coherence during speed instructions) contributed to the average. Thereby we ensured the inclusion of at least 50% of trials at all considered time points for all conditions. For statistical analysis, we identified the time window in which beta showed a cue-induced decrease (*Oswal et al., 2012*) based on the average across all conditions (see *Figure 4D*) and computed the change in beta between the start of the beta decrease (150 ± 25 ms post-cue) and the end of the beta decrease (400 ± 25 ms post-cue) for each patient and condition. Importantly, this time period was well before the average RT (~1250 ms post-cue). We then conducted a 2*2 ANOVA with instruction and coherence as factors after ensuring that assumptions of normality were met using Shapiro-Wilk test.

Since we found an effect of instruction on LFO power both for response- and cue-aligned spectra (see Results) we additionally analyzed whether changes in LFO power differed between alignments by extracting the mean LFO power from the relevant windows (−750 ms before the response until the response vs. 500 ms to 1250 ms postcue) for speed and accuracy instructions separately and conducted a 2*2 ANOVA with factors alignment (response vs. cue) and instruction (speed vs.

accuracy). Finally, we also computed the intertrial phase clustering (ITPC) for STN LFO in order to test whether the LFO phase during the above mentioned windows of interest was time-locked to the cue or response. To this end, we projected the phase at each time point of each trial onto the complex plane, averaged across trials and computed its absolute value (*Cohen, 2014a*; *Zavala et al., 2013*, *2016*), as follows:

$$ITPC(t) = \left| \frac{1}{N} \sum_{n=1}^{N} e^{i\phi_{n,t}} \right|$$

where $\phi_{n,t}$ is the phase angle at trial n and time t. We computed the mean LFO ITPC for the above mentioned 750 ms time windows for the cue- and response-aligned data after speed and accuracy instructions separately. The resulting ITPC is bound between 0 and 1, where 0 indicates phase inconsistency across trials and one means that the phase at a given time point is identical for each trial. These values were then averaged across participants and compared against a critical ITPC value given by

$$ITPC_{critical} = \sqrt{\frac{-\ln(p)}{n}}$$

where n is the average number of trials (201) and p the critical p-value of 0.05 (*Cohen, 2014a*).

## Drift diffusion model

In the DDM framework, perceptual decision-making between two alternatives is reflected by a continuous integration of sensory evidence over time until sufficient evidence has been accumulated and the choice is executed. DDM has been widely applied over the last decades and has been shown to accurately predict behavior over a range of different tasks (*Ratcliff and McKoon, 2008*). There are three main parameters in DDM. First, the drift rate *v* reflects the rate of evidence accumulation. If a cue clearly favors one over the other choice the drift rate is high resulting in fast and accurate decisions, while ambiguous cues will lead to low drift rates and thus slow and error-prone choices. In our experiment, we expected the level of dots coherence to modulate the drift rate (*Ratcliff and McKoon, 2008*). Second, the decision threshold *a* defines how much evidence is accumulated before committing to a choice. Thus, the decision threshold constitutes a decision criterion, which transforms a continuous variable (sensory evidence) into a categorical choice (option A or B). There is ample evidence that speed-accuracy adjustments are mediated through modulations of the decision threshold so that decreased thresholds (or mathematically equivalent an increased baseline, see *Figure 2A*) during speed emphasis lead to faster responses at the expense of accuracy. This leads to increased fast errors during speed emphasis, that is, errors are typically faster than correct trials (*Ratcliff and McKoon, 2008*). Conversely, during conditions with high thresholds errors are typically slower than correct trials (for a more detailed explanation of this finding see [*Ratcliff and Rouder, 1998*]). In the current study, we hypothesized that the decision threshold would be modulated by speed vs. accuracy instructions. The third parameter in DDM is the non-decision time *t*, which is thought to be related to afferent delay, sensory processing and motor execution.

We applied a Bayesian hierarchical estimation of DDM (HDDM), which is particularly suited for experiments with low trial counts (*Wiecki et al., 2013*), implemented in Python 2.7.10 (RRID:SCR_008394). Another advantage of HDDM is that it allows regression analyses between trial-by-trial variations of model parameters and fluctuations of neural activity, such as STN power modulations. In other words, model parameters and neural activity are not analyzed separately and then correlated afterwards (which would result in n = 11 observations (patients) in the current study), but the neural variables are directly entered into the model and allowed to modulate the latent decision-making parameters at each trial (resulting in n = 2209 observations (trials) in our study). The hierarchical design assumes that parameters from individual participants are not completely independent, but drawn from the group distribution while allowing variations from this distribution given sufficient evidence to overwhelm the group prior. Prior distributions were informed by 23 previous studies (*Wiecki et al., 2013*). The starting parameter (bias parameter) *z* was fixed to 0.5, because leftwards and rightwards movements were equally likely. We *a-priori* assumed that the decision threshold *a* was modulated by instruction (speed vs. accuracy) and the drift rate *v* by level of coherence (low vs.

high) allowing for overall trial-by-trial variability in drift. In addition, we also assessed whether the non-decision time $t$ was affected by changes in instructions or dots coherence. Since this was not the case (see Results), we used a simple model with drift rates affected by coherence and thresholds altered by instructions for model checks and regression analyses (see below). Markov chain Monte Carlo sampling was used for Bayesian approximation of the posterior distribution of model parameters generating 20,000 samples and discarding 10,000 samples as burn-in (*Herz et al., 2016*). To ensure model convergence we inspected traces of model parameters, their autocorrelation and computed the R-hat (Gelman-Rubin) statistics (*Wiecki et al., 2013*). To assess model performance, we computed quantile probability plots (*Ratcliff and McKoon, 2008*), in which predicted and observed RT for the 10, 30, 50, 70 and 90 percentile were plotted against their predicted and observed cumulative probability for each condition. Error trials were only plotted for the low coherence condition due to the paucity of errors in the high coherence condition (~5%). Parameters of the model were analyzed by Bayesian hypothesis testing. We considered posterior probabilities ≥95% of the respective parameters being different than zero significant (*Frank et al., 2015*; *Herz et al., 2016*). In other words, model parameters were significant if ≥95% of samples drawn from the posterior were different from zero (or different from the distribution they were compared to, for example, of healthy participants). Even though such *posterior probabilities* are distinct from classical p values (e.g. in a t-test) they can be interpreted in a similar manner (*Wiecki et al., 2013*). After having conducted the HDDM analysis for patients, we conducted the identical analysis for healthy participants to assess whether task effects on latent decision-making parameters were different between both groups.

## HDDM with single-trial STN activity

After estimating the HDDM not containing any neural data, we entered neural variables, which were related to speed-accuracy adjustments as observed in the analysis of electrophysiological data, into the model. In particular, we fitted a model assuming that a threshold on a given trial depended on measured neurophysiological data. These data comprised LFO power in the pre-response period and the cue-induced decrease in beta power (see Results). Based on the trial-averaged group comparison, single trial LFO power was extracted from −750 ms until the response or from –RT until the response in case RT were <750 ms (*Herz et al., 2016*), that is, locked to the response. The beta-power decrease was extracted using the change from 150 ms to 400 ms (±25 ms) after cue-onset (see above). Then single trial values were averaged across the respective time window and frequencies. Trials with RT <425 ms (1.7%) were excluded from the HDDM analysis to ensure that the dots-induced beta decrease did not fall into the time of the motor response and the remaining single trial values were z-scored by subtracting the mean and dividing by the standard deviation for each subject. We assessed a putative statistical relationship between the cue-induced beta decrease and pre-response LFO increase, by conducting Spearman correlations for each subject, Fisher r-to-z-transforming the correlation coefficients and then testing the resulting values against 0 using a one-sample t-test on the second level. Since we hypothesized that the changes in STN activity observed during speed-accuracy adjustments were related to modulations of decision thresholds, we regressed these neural variables against estimates of thresholds at each trial during model estimation. In other words, the regression coefficients between STN activity and decision thresholds were estimated within the same model, which was used to estimate the decision-making parameters themselves. For example, on a given trial the threshold $a$ was defined by:

$a = b_0 + b_1Instr + b_2LFO + b_3LFO*Instr + b_4Betadecrease + b_5Betadecrease*Instr$, where *Instr* refers to the type of Instruction (0 for Accuracy, one for Speed), *LFO* indicates the pre-response increase in LFO, *Betadecrease* the cue-induced decrease in beta power, and $b_{1-5}$ are the estimated regression coefficients.

Of note, since we included single-trial power in LFO and beta in the same model, we accounted for effects of beta on thresholds for the LFO regression estimates and vice versa. Even though we hypothesized that changes in STN power were related to modulations of decision thresholds, we additionally computed an identical regression analysis with drift rate estimates as dependent variable in the same model as a control analysis. Posteriors of regression coefficients for trial-wise regressors were estimated only at the group level to address potential collinearity among model parameters, for regularizing parameter estimates and to prevent parameter explosion (*Frank et al., 2015*; *Wiecki et al., 2013*). Statistical inference on regression coefficients was based on the distribution of the posterior probability densities (see above). Finally, we computed the deviance

information criterion (DIC) for the HDDM not containing any neural data and the HDDM containing neural activity in order to assess whether including STN activity improved model performance. DIC is mainly used for comparisons of hierarchical models where other measures, such as the Bayesian information criterion (BIC), are not appropriate (*Frank et al., 2015*; *Wiecki et al., 2013*). Lower DIC values indicate improved model fits taking into account model complexity. Traditionally, DIC differences >10 are considered significant (*Herz et al., 2016*).

In order to assess whether the observed relationships between STN activity and HDDM parameters (see Results) were also evident in simple behavioral measures, we applied additional regression analyses using STN power (LFO and beta) as predictors and RT as dependent variable. We used a linear mixed-effects regression model, in which the intercept of the regression was allowed to vary between participants (random effect), while the slope of the regressions was treated as fixed effect (consistent across subjects), similar to the HDDM regression analysis described above. Since we tested specific directional effects based on the HDDM regression analyses, for example, stronger cue-induced beta-decreases predicted lower thresholds (see Results), we used $P_{one-tailed} = 0.05$ as statistical threshold.

## Statistical analysis of EEG data and cortico-STN connectivity

EEG was recorded in addition to STN LFPs for computation of cortico-STN connectivity. More specifically, we wanted to assess whether modulations of STN activity in the LFO and beta band were related to changes in Fz-STN connectivity and C3/C4-STN connectivity, respectively, and whether this relationship was modulated by speed vs. accuracy instructions. In a first step, we plotted response- and cue-aligned spectra for Fz and C3/C4 separately. To analyze whether task-related changes in LFO and beta were predominantly expressed over Fz or C3/C4, we compared the pre-response increase in LFO and cue-induced decrease in beta (see Statistical analysis of STN LFPs) between these two sites using permutation testing (1000 permutations) by shuffling between electrode labels for each subject.

After having established the spatial selectivity of cortical LFO and beta activity, we then computed the inter-site phase clustering (ISPC) between Fz and STN in the LFO band as well as between C3/C4 and STN in the beta band. ISPC is a phase-based measure of connectivity reflecting to what extent oscillations in two different areas are phase-locked at specific time points (*Cohen, 2014a*; *Herz et al., 2016*; *Zavala et al., 2013*, *2014*, *2016*). This measure is independent of power changes in that it only considers the phase information of the signal and will be only affected by power in cases where power is very low (because phase estimates deteriorate) or zero (because phase cannot be estimated with zero power) (*Cohen, 2014a*). Note that we did not compute single trial estimates of connectivity, because connectivity estimates are far more robust when averaging across trials. After extracting the phase from the wavelet transform, the difference between the phase of the STN LFP and respective EEG channel was calculated at each time point of each trial, and averaged across trials. Note that this computation is identical to ITPC described above, except that the phase difference between the STN and EEG signal were used rather than phase angles from STN alone. The resulting values were then integrated over time using a sliding window decreasing linearly from ±200 ms (corresponding to ±1 cycle for 5 Hz, the median frequency of LFO) to ±100 ms (roughly corresponding to ±2 cycles for 22 Hz, the median of beta). For specifically assessing task-related changes in ISPC, we calculated the % change from baseline. The baseline was defined as the time from 500 ms until 250 ms before onset of the moving dots cue. The time window of interest was defined based on the modulations of STN power during speed-accuracy adjustments: a 750 ms time window prior to the response for LFO and a 500 ms time window starting 150 ms after dots onset for beta. For LFO, this was computed for Fz-left STN and Fz-right STN and then averaged across hemispheres. For beta, C3-left STN was used for right-handed responses and C4-right STN for left-handed responses before averaging across hemispheres. This resulted in one ISPC value per patient (n = 8) for each cortico-STN connection (Fz-STN and C3/C4-STN). To assess putative effects of speed-accuracy adjustments, we then compared ISPC for Fz-STN and C3/C4-STN between speed and accuracy instructions. We applied permutation testing by shuffling between condition labels using 1000 permutations. Finally, we assessed whether differences in STN power, which were shown to predict trial-by-trial adjustments of decision threshold (see Results), were related to differences in cortico-STN connectivity across subjects and whether this relationship depended on the type of instruction. To this end, we entered the ISPC values into a Pearson correlation analysis with the

corresponding STN power (i.e. pre-response STN LFO power for Fz-STN ISPC and cue-induced STN beta power decrease for C3/C4-STN ISPC) separately for speed and accuracy instructions. The resulting rho values for speed and accuracy were Fisher r-to-z transformed and compared between the speed and accuracy condition. In case of a significant difference the correlations within a condition, that is, during accuracy and speed conditions separately, were tested for significance. Prior to conducting Pearson correlations, we ensured that assumptions of parametric correlations were met including absence of outliers using Grubbs test and normality using Shaprio-Wilk test. Please note that single subject estimates of decision thresholds were not used for any correlation analyses, since the hierarchical design of HDDM violates the assumption of independence of observations (*Wiecki et al., 2013*). For all statistical tests of EEG data and cortico-STN connectivity, a statistical threshold of p=0.05 was applied correcting for multiple comparisons using the Bonferroni method whenever applicable. For permutation tests, p-values are given as p<0.05, p<0.01 or p<0.001 instead of precise p-values, since these can change slightly when repeating permutation tests.

## Acknowledgements

This project has received funding from the Medical Research Council (award MC_UU_12024/1) and the European Union's Horizon 2020 research and innovation programme under the Marie Sklodowska-Curie grant agreement No 655605. The Unit of Functional Neurosurgery is supported by Parkinson Appeal UK and the Monument Trust. The behavioral and neurophysiological data is freely available. doi: 10.5287/bodleian:VEd27b0Yr

## Additional information

### Funding

| Funder | Grant reference number | Author |
|---|---|---|
| Medical Research Council | MC_UU_12024/1 | Peter Brown |
| Horizon 2020 Framework Programme | 655605 | Damian M Herz |
| Parkinson Appeal UK | | Thomas Foltynie<br>Patricia Limousin<br>Ludvic Zrinzo |
| Monument Trust | | Thomas Foltynie<br>Patricia Limousin<br>Ludvic Zrinzo |

The funders had no role in study design, data collection and interpretation, or the decision to submit the work for publication.

### Author contributions

DMH, Conceptualization, Data curation, Formal analysis, Funding acquisition, Methodology, Writing—original draft, Project administration, Writing—review and editing; HT, Data curation, Formal analysis, Methodology, Writing—review and editing; J-SB, Formal analysis, Methodology, Writing—review and editing; PF, Data curation, Methodology, Writing—review and editing; BC, ALG, JF, TZA, KA, SL, TF, PL, LZ, Data curation, Writing—review and editing; RB, Conceptualization, Formal analysis, Supervision, Methodology, Writing—review and editing; PB, Conceptualization, Resources, Data curation, Formal analysis, Supervision, Funding acquisition, Methodology, Project administration, Writing—review and editing

### Author ORCIDs

Huiling Tan, http://orcid.org/0000-0001-8038-3029
Petra Fischer, http://orcid.org/0000-0001-5585-8977
Peter Brown, http://orcid.org/0000-0002-5201-3044

## Ethics

Human subjects: In accordance with the declaration of Helsinki, participants gave written informed consent to participate in the study, which was approved by the local ethics committee (Oxfordshire REC A).

## Additional files

### Supplementary files

• Supplementary file 1. Clinical specifications. Age and disease duration are given in years. UPDRS-III: Unified Parkinson's disease rating scale part III.

### Major datasets

The following dataset was generated:

| Author(s) | Year | Dataset title | Dataset URL | Database, license, and accessibility information |
|---|---|---|---|---|
| Damian Herz | 2017 | Neural correlates of speed-accuracy adjustments in the subthalamic nucleus | https://doi.org/10.5287/bodleian:VEd27b0Yr | Publicly available at Oxford University Research Archive (uuid: 09bef38c-999f-4fb7-aa46-14eda3123571) |

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
