## [Decision Letter]

Thank you for submitting your article "Distinct mechanisms mediate speed-accuracy adjustments in cortico-subthalamic networks" for consideration by *eLife*. Your article has been favorably evaluated by Sabine Kastner (Senior Editor) and three reviewers, one of whom, Peter Lakatos (Reviewer #1), is a member of our Board of Reviewing Editors and another one is Jonas Obleser (Reviewer #2).

The reviewers have discussed the reviews with one another and the Reviewing Editor has assembled a list of concerns, the first in particular, which require your attention. Before committing to a binding decision, we ask you to respond to the issues raised with a plan to address the points discussed below, particularly the concern about a control group of normal patients evaluated by concurrent EEG recordings or a behavioral control experiment. It would help to know approximately how much time you would require to address these concerns.

Summary:

The study by Hertz et al. was aimed at revealing the neurophysiological correlates of the speed-accuracy tradeoff aspect of decision making by recording local field potentials (LFPs) directly from the subthalamic nucleus (STN) of Parkinson's disease patients with concurrent electroencephalography (EEG). The behavioral data and computational modeling of three of the subjects' decision making parameters support the notion that subjects adjusted their decision thresholds based on whether they were instructed to emphasize speed vs. accuracy during the performance of a moving dots task. The authors also demonstrate two electrophysiological correlates of speed vs. accuracy emphasis in STN: a low frequency oscillation (LFO) amplitude increase and β decrease in "speed trials" trials compared to "accuracy trials". Further, they show that while the LFO effect dominates EEG recordings above PFC, the β effect dominates EEG recordings above motor cortex, and that these are coherent with STN oscillatory activity that corresponds in frequency and timing, possibly indicating the involvement of distinct networks. These findings will be important for future studies aimed at revealing the circuitry and spectrotemporal fingerprints of perceptual-cognitive operations, since they emphasize that the brain might prioritize different mechanisms and underlying networks based on how subjects interpret task requirements.

Essential revisions:

1) An important concern raised by one of the reviewers is that since motor-cognitive behavior is altered in Parkinson's patients, the authors should replicate their experiments in a control group of healthy, age-matched controls, to show that their performance trends are similar to patients'. Ideally, concurrent EEG recordings would further validate the results obtained from patients.

2) Since the missing accuracy effect could in theory be caused by a concomitant change in bias and sensitivity, a more thorough analysis of the behavioral data in terms of signal detection is also warranted (hit rate, false alarms).

3) Overall, all reviewers noted that due to methodological limitations of the electrophysiological recordings, some of the claims related to specific networks should be toned down. Specifically, when referring to EEG recordings, throughout the Results the authors should use the electrode labels (C3/C4 and F3) instead of "motor" or "prefrontal cortex", and reserve the claim about PFC – STN, and M1 – STN for interpretation. Likewise, the important caveat needs to be mentioned that while STN LFP recordings are indisputably local, they cannot show specificity to the STN. It is possible that similar signatures of β and LFO changes would be evident in many or all basal ganglia nuclei if it were possible to record from them, c.f. Leventhal et al. 2012, Neuron. This should also lead to moderation of claims in the abstract about "prefrontal – STN' and "M1- STN" networks. To summarize, the results should be reported in more neutral terms, with the interpretation (based on the wider background literature and theory) that these signals reflect these specific networks, but further work ahead.

4) The reviewers would also like to see the study better framed by considering/citing classic studies on the speed-accuracy tradeoff (prior to the 90's), and a discussion on how the current results on STN LFOs might/might not align with previous STN LFO studies comparing high vs. low 'conflict' scenarios (e.g. Zavala et al. 2014 J Neuroscience, Brittain et al. 2012 J Neuroscience, and papers by Cavanagh and others).

5) It is unclear how much we can make of altered phase coupling if one of the contributing variables is also changed in power simultaneously, therefore, the authors are urged to include a cautionary note when interpreting their phase measures.

6) The (plausible) reasoning in the Discussion that "Since responses occurred much earlier in high vs. low coherence trials than in speed vs. accuracy trials, it is unlikely that the more pronounced β decrease during speed emphasis […] simply reflects the likelihood of a movement" should be substantiated by a control analysis (supplement-level), where β decrease binned by response time is shown. Similarly, the claim in the Results, and Figure 6: "While EEG over PFC [I suggest reads 'F3'] showed little modulation in the β band, but pronounced changes in the LFP band [Figure 6], activity over M1 [I suggest reads 'C3/C4'] displayed the opposite pattern, namely a strong β modulation, but no clear changes in the LFO band [Figure 6]" should be substantiated by statistical analyses.

7) While all reviewers were impressed by the combination of behavioral data/electrophysiological recordings and the sophisticated drift diffusion modeling, they would also like to see how LFO/β power relate to simple behavioral parameters like reaction time and accuracy.

8) One of the reviewers thought that it would be of benefit to the study if the authors would further analyze the relationship of stimulus onset and oscillations, especially LFOs. The current results seem to indicate that pre-response, stimulus onset (cue) aligned effects are greater both for low-frequency oscillatory (LFOs) and β oscillatory amplitudes (Figure 3–Figure 4), which could be statistically tested. Additionally, calculating the inter-trial coherence of low-frequency oscillations related to stimulus onset vs. response onset might reveal which of these two events LFOs are most synchronized to, indicating possible phase reset by stimulus onset or behavioral responding. If it turns out that LFOs are synchronized to stimulus onset (significant inter-trial coherence), and that this synchronization (ITC) is stronger in the case when speed is emphasized over accuracy, one could interpret this as in this case, bottom-up processes like oscillatory phase reset become more heavily involved in decision making.

9) What timeframe and frequency range were β and LFO amplitudes measured for the single trial analyses shown in Figure 5 and elsewhere (e.g. EEG LFO and β statistics described in the first paragraph of the subsection “Modulation of cortico-STN connectivity during speed-accuracy adjustments”)? Do the terms "pre-response LFO power" and "cue-induced β power" indicate that while LFO amplitudes were extracted from response aligned, β amplitudes were extracted from cue aligned trials? If the above suggested analyses indicate that LFO modulations are stronger and LFOs are "more phase-locked" in stimulus onset aligned data, both LFO and β amplitudes should be measured in timeframes aligned to stimulus onset. This might reveal a correlation between the two variables (Figure 5).

10) Lastly, one of the reviewers noted that while Abstract says that "… while cue-induced reductions of subthalamic β power decreased thresholds irrespective of instructions, but were more *pronounced* when speed was emphasized." the results apparently show no difference, i.e. in the subsection “Combining recorded STN power changes with drift diffusion modeling”, "… the relationship between β power decreases and decision thresholds did not depend on speed vs. accuracy instructions", and see Figure 5. This appears discrepant.

---

## [Author Response]

*Essential revisions:*

*1) An important concern raised by one of the reviewers is that since motor-cognitive behavior is altered in Parkinson's patients, the authors should replicate their experiments in a control group of healthy, age-matched controls, to show that their performance trends are similar to patients'. Ideally, concurrent EEG recordings would further validate the results obtained from patients.*

We agree that it is important to ensure that the patients enrolled in the current study were able to perform the task as would be expected from healthy participants. We therefore followed the reviewers’ suggestion to conduct an additional experiment in an age-matched control group (n=18). This experiment replicated the task effects on reaction times (RT) and accuracy rates, which we had observed in Parkinson’s disease patients (see subsection “Behavioral control experiment in healthy participants”, first paragraph and Figure 1—figure supplement 1). Furthermore, we conducted drift diffusion modelling of the data from healthy participants and confirmed that decision threshold estimates did not differ between patients and healthy controls and that both groups had increased decision thresholds after accuracy compared to speed instructions in line with our a-priori hypothesis (see the second paragraph of the aforementioned subsection and Figure 2—figure supplement 1).

*2) Since the missing accuracy effect could in theory be caused by a concomitant change in bias and sensitivity, a more thorough analysis of the behavioral data in terms of signal detection is also warranted (hit rate, false alarms).*

We agree that accuracy rates are only a coarse measure of performance. This is particularly true when they are analysed independently from RT, since both measures are closely related to each other (the likelihood of a response to be correct depends on the time taken for deliberation). However, our task design is unfortunately not well suited for signal detection analyses. This is because in our paradigm there is always a stimulus (moving dots) present, which in some trials is easier to detect (high coherence) than in others (low coherence), but there are no trials with only randomly moving dots. In other words, there is always a direction in which the majority of dots move. Therefore, there are no correct response omissions and, similarly, the error of omission rate is very low (< 2%) rendering signal detection analyses unreliable. On the other hand, drift diffusion modelling can be interpreted as a related or extended approach to signal detection theory, which does not only take into account accuracy rates, but also the complete RT distribution of the behavioural data (for an interesting discussion of signal detection theory and sequential analysis / drift diffusion modelling see Gold and Shadlen, 2007, Annual Review of Neuroscience). Drift diffusion analysis indeed confirms what the reviewers indicated; a significant effect on a latent parameter underlying the observed behaviour (in this case decision thresholds), which could not be observed when only analysing accuracy rates. In addition, Figure 1 shows that errors were primarily observed during slow responses only after accuracy, but not speed instructions; a finding, which has been related to conditions with high decision thresholds (see Ratcliff & McKoon, 2008, Neural Computation). We hope that these observations further support the notion that task instruction indeed altered task performance even in the absence of an effect on accuracy rates when analysing them separately from RT.

*3) Overall, all reviewers noted that due to methodological limitations of the electrophysiological recordings, some of the claims related to specific networks should be toned down. Specifically, when referring to EEG recordings, throughout the Results the authors should use the electrode labels (C3/C4 and F3) instead of "motor" or "prefrontal cortex", and reserve the claim about PFC – STN, and M1 – STN for interpretation. Likewise, the important caveat needs to be mentioned that while STN LFP recordings are indisputably local, they cannot show specificity to the STN. It is possible that similar signatures of β and LFO changes would be evident in many or all basal ganglia nuclei if it were possible to record from them, c.f. Leventhal et al. 2012, Neuron. This should also lead to moderation of claims in the abstract about "prefrontal – STN' and "M1- STN" networks. To summarize, the results should be reported in more neutral terms, with the interpretation (based on the wider background literature and theory) that these signals reflect these specific networks, but further work ahead.*

We have followed the reviewers’ suggestions and now refer to cortico-STN connectivity using the respective electrode labels rather than referring to the putative origin (i.e. prefrontal and motor) of the recorded signals. In the Discussion we have also incorporated a more extensive summary of previous findings regarding the topography of cortical low frequency oscillatory (LFO) and β power, which facilitate interpretation of the findings in the current study (Discussion, third paragraph). Finally, we now explicitly mention that the findings of threshold-related signals in the STN do not imply that within the basal ganglia only the STN is involved in thresholds adjustments, but that there might well exist more distributed activity changes (subsection “Analysis of behavioral data”).

*4) The reviewers would also like to see the study better framed by considering/citing classic studies on the speed-accuracy tradeoff (prior to the 90's), and a discussion on how the current results on STN LFOs might/might not align with previous STN LFO studies comparing high vs. low 'conflict' scenarios (e.g. Zavala et al. 2014 J Neuroscience, Brittain et al. 2012 J Neuroscience, and papers by Cavanagh and others).*

We now refer to earlier (pre 90s) studies on the speed-accuracy tradeoff in the Introduction and refer to a review on the historical developments in speed-accuracy tradeoff research for the interested reader (Introduction). In addition, we included a paragraph discussing how the results of the current study relate to previous studies assessing the role of the STN and prefrontal cortex during response ‘conflict’ (in particular Zavala et al., Brittain et al. & Cavanagh et al.) in the second paragraph of the Discussion.

*5) It is unclear how much we can make of altered phase coupling if one of the contributing variables is also changed in power simultaneously, therefore, the authors are urged to include a cautionary note when interpreting their phase measures.*

We have included a cautionary note in the Discussion mentioning how power changes (in particular low power) can affect phase estimates and thus phase-based connectivity (third paragraph).

*6) The (plausible) reasoning in the Discussion that "Since responses occurred much earlier in high vs. low coherence trials than in speed vs. accuracy trials, it is unlikely that the more pronounced β decrease during speed emphasis […] simply reflects the likelihood of a movement" should be substantiated by a control analysis (supplement-level), where β decrease binned by response time is shown. Similarly, the claim in the Results, and Figure 6: "While EEG over PFC [I suggest reads 'F3'] showed little modulation in the β band, but pronounced changes in the LFP band [Figure 6], activity over M1 [I suggest reads 'C3/C4'] displayed the opposite pattern, namely a strong β modulation, but no clear changes in the LFO band [Figure 6]" should be substantiated by statistical analyses.*

We realized that the statement regarding the effects of RT differences on the cue-induced β decrease was unclear in the previous version of the manuscript. We were referring to the absence of an effect of coherence on the cue-induced β decrease even though low and high coherence trials had much stronger RT differences than speed vs. accuracy trials. However, the regression analysis showing a significant negative relationship between the β-decrease and decision thresholds in fact shows that this LFP change is related to RT differences. Furthermore, we now show that the cue-induced β decrease is also negatively correlated with RT (see reply to comment 7). Therefore, we believe that the above statement in the previous version of the manuscript was misleading and have removed it from the article. Regarding statistical analysis of EEG data, we have now restructured this part of the Results sections, so that the statistical analysis is shown in the same sentence stating that LFO changes were stronger in Fz compared to C3/C4, while the opposite was the case for β changes (subsection “Modulation of cortico-STN connectivity during speed-accuracy adjustment”).

*7) While all reviewers were impressed by the combination of behavioral data/electrophysiological recordings and the sophisticated drift diffusion modeling, they would also like to see how LFO/β power relate to simple behavioral parameters like reaction time and accuracy.*

We would like to thank the reviewers for this helpful comment and agree that testing whether there are similar effects of STN activity on simple behavioural measures further strengthens the results obtained using computational modelling. We have now conducted additional regression analyses using RT rather than thresholds as dependent variable, which demonstrate that similar effects of STN power changes can be detected. More specifically, pre-response LFO increases predict longer RT only after accuracy, but not speed instructions, while stronger cue-induced β power decreases predict shorter RT across instructions (subsection “Combining recorded STN power changes with drift diffusion modeling”, second paragraph). Regarding accuracy rates, we only have limited LFP data for incorrect trials, since error rates were < 25% in low coherence trials (and virtually 0% in high coherence trials), so that only approximately 10% of all trials are errors. Therefore, we did not attempt any regression analyses (e.g. logistic regression). However, we compared STN time-frequency spectra between correct and error trials, which did not reveal any differences (see Figure 3—figure supplement 1).

*8) One of the reviewers thought that it would be of benefit to the study if the authors would further analyze the relationship of stimulus onset and oscillations, especially LFOs. The current results seem to indicate that pre-response, stimulus onset (cue) aligned effects are greater both for low-frequency oscillatory (LFOs) and β oscillatory amplitudes (Figure 3–Figure 4), which could be statistically tested. Additionally, calculating the inter-trial coherence of low-frequency oscillations related to stimulus onset vs. response onset might reveal which of these two events LFOs are most synchronized to, indicating possible phase reset by stimulus onset or behavioral responding. If it turns out that LFOs are synchronized to stimulus onset (significant inter-trial coherence), and that this synchronization (ITC) is stronger in the case when speed is emphasized over accuracy, one could interpret this as in this case, bottom-up processes like oscillatory phase reset become more heavily involved in decision making.*

We followed the reviewer’s suggestion and directly compared LFO power aligned to the cue vs. aligned to the response using 750 ms windows (i.e. 750 ms before the response until the response, which was also used for correlation and regression analyses in all following analyses, vs. 500 ms until 1250 ms after cue onset). This analysis did not reveal any significant differences between the response- and cue-aligned data (subsection “Changes in STN activity during speed-accuracy adjustments”, second paragraph). Furthermore, we computed the inter-trial coherence (referred to as intertrial phase clustering). This showed that neither the response-, nor the cue-aligned changes in LFO showed any significant phase-locking across trials in the respective time windows arguing against a strong evoked component (in the fourth paragraph of the aforementioned subsection). We therefore continue to use the response-related changes in LFO (750 ms before the response until the response) for all further analyses as in the previous version of the paper. This has the advantage compared to cue-aligned data that the single trial LFO changes will not fall into the post-response period for any of the trials and is consistent with the approach used in a previous paper assessing the relationship between LFO power and decision thresholds (Herz et al., 2016, Current Biology).

9) What timeframe and frequency range were β and LFO amplitudes measured for the single trial analyses shown in Figure 5 and elsewhere (e.g. EEG LFO and β statistics described in the first paragraph of the subsection “Modulation of cortico-STN connectivity during speed-accuracy adjustments”)? Do the terms "pre-response LFO power" and "cue-induced β power" indicate that while LFO amplitudes were extracted from response aligned, β amplitudes were extracted from cue aligned trials? If the above suggested analyses indicate that LFO modulations are stronger and LFOs are "more phase-locked" in stimulus onset aligned data, both LFO and β amplitudes should be measured in timeframes aligned to stimulus onset. This might reveal a correlation between the two variables (Figure 5).

LFO power was extracted from 750ms before the response until the response, while changes in β power were extracted from 150 ms to 400 ms after the cue based on the trial-averaged spectra shown in Figure 3 and 4. We have now made this more clear in the main text (subsection “Changes in STN activity during speed-accuracy adjustments”, second paragraph). Please see reply to comment 8 regarding strengths and phase-locking of cue- vs. response-aligned LFO.

*10) Lastly, one of the reviewers noted that while Abstract says that "… while cue-induced reductions of subthalamic β power decreased thresholds irrespective of instructions, but were more pronounced when speed was emphasized." the results apparently show no difference, i.e. in the subsection “Combining recorded STN power changes with drift diffusion modeling”, "… the relationship between β power decreases and decision thresholds did not depend on speed vs. accuracy instructions", and see Figure 5. This appears discrepant.*

We apologize for the unclear statement in the previous version of the manuscript. It was supposed to say that the β power decrease was related to decreased thresholds irrespective of instruction, while the decrease in β power (not its relationship with thresholds) was more pronounced after speed instructions. We have re-written the Abstract to avoid this misunderstanding.